# Improving Deep Learning Speed and Performance through Synaptic Neural Balance

**Antonios Alexos**
Department of Computer Science
University of California Irvine
Irvine, CA 92697
aalexos@uci.edu

**Ian Domingo**
Department of Computer Science
University of California Irvine
Irvine, CA 92697
idomingo@uci.edu

**Pierre Baldi**
Department of Computer Science
University of California Irvine
Irvine, CA 92697
pfbaldi@uci.edu

## Abstract

We present experiments and their corresponding theory, demonstrating that synaptic neural balancing can significantly enhance deep learning speed, accuracy, and generalization, particularly on non-traditional compute paradigms. Given an additive cost function (regularizer) of the synaptic weights, a neuron is in balance if the total cost of its incoming weights equals that of its outgoing weights. For various networks, activation functions, and regularizers, neurons can be balanced using scaling operations without altering their functionality, associated with a strictly convex optimization problem. In our simulations, we systematically observe that: (1) Fully balancing before training results in better performance as compared to several other training approaches; (2) Interleaving partial (layer-wise) balancing and stochastic gradient descent steps during training results in faster learning convergence and better overall accuracy (with $L_1$ balancing converging faster than $L_2$ balancing; and (3) When given limited training data, neural balanced models outperform plain or regularized models. and this is true both for both feedforward and recurrent networks. These balancing operations are entirely local, making them viable for biological or neuromorphic systems. This positions synaptic neural balancing as a promising approach for leveraging the unique characteristics of emerging AI accelerators, advancing the efficiency and sustainability of machine learning.

## 1 Introduction

Broadly speaking, neural balance refers to the idea of achieving or keeping a certain equilibrium in a neural network during training or after training, whereby such equilibrium may facilitate better information flow, or lower energy expenditure Shwartz-Ziv [2022]. As such, there are different notions of neural balance including, for example, the notion of balance between excitation and inhibition in biological neural networks [Froemke, 2015, Field et al., 2020, Howes and Shatalina, 2022, Kim and Lee, 2022, Shirani and Choi, 2023]. Here we develop the concept of synaptic neural balance which refers to any systematic relationship between the input and output synaptic weights of individual neurons, or layers of neurons. Specifically, we consider the case where the cost of the input weights is equal to the cost of the output weights, where the cost is defined by some regularizer. One of the most basic examples of such a relationship, described below, is when the sum of the squares of

38th Second Workshop on Machine Learning with New Compute Paradigms at NeurIPS 2024(MLNCP 2024).

the input weights of a neuron is equal to the sum of the squares of its output weights. In this work, we briefly describe the theory of synaptic neural balance and demonstrate its applications to deep learning regularization. We now describe the base case of synaptic neural balance.

**Base Case:** Consider a neuron with a ReLU activation function inside a network trained to minimize a regularized error function $\mathcal{E} = E + R$, where $E$ is the data-dependent error (typically the negative log-likelihood of the data) and $R$ is the regularizer (typically $L_2$ regularizer). If we multiply the incoming weights of the neuron by some $\lambda > 0$ (including the bias) and divide the outgoing weights of the neuron by the same $\lambda$, it is easy to see that this scaling operation does not affect in any way the contribution of the neuron to the rest of the network. Thus, the error $E$ which depends only on the input-output function of the network is unchanged. However, the value of the $L_2$ regularizer changes continuously with $\lambda$, and the corresponding contribution is given by:

$$\sum_{i \in IN} (\lambda w_i)^2 + \sum_{i \in OUT} (w_i/\lambda)^2 = \lambda^2 A + \frac{1}{\lambda^2} B \qquad (1)$$

where $IN$ and $OUT$ denote the set of incoming and outgoing weights respectively, $A = \sum_{i \in IN} w_i^2$, and $B = \sum_{i \in OUT} w_i^2$. When $\lambda$ moves away from 1, the contribution increases in one direction and decreases in the other. In the direction where it decreases, we can solve for the value $\lambda^*$ associated with the mimimal cost. Without taking derivatives, we note that the product of the two terms on the right-hand side of Equation 1 is equal to $AB$ and does not depend on $\lambda$. Thus, the minimum is achieved when these two terms are equal, which yields: $(\lambda^*)^4 = B/A$ for the optimal $\lambda^*$. The corresponding new set of weights, $v_i = \lambda^* w_i$ for the input weights and $v_i = w_i/\lambda^*$ for the outgoing weights, must be balanced: $\sum_{i \in IN} v_i^2 = \sum_{i \in OUT} v_i^2$. This is because the optimal scaling factor for the optimal synaptic weights can only be $\lambda^* = 1$. Thus, we can define two operations that can be applied to the incoming and outgoing weights of a neuron: scaling and balancing. In between, we can also consider favorable scaling, or partial balancing, where $\lambda$ is chosen to reduce the cost without necessarily minimizing it.

There have been isolated previous studies of this kind of synaptic balance [Du et al., 2018, Stock et al., 2022] under special conditions. For instance, in Du et al. [2018], it is shown that if a deep network is initialized in a balanced state with respect to the sum of squares metric, and if training progresses with an infinitesimal learning rate, then balance is preserved throughout training. However, using an infinitesimal learning rate is not practical. Furthermore, there are many intriguing questions that can be raised. For instance: Why does balance occur? Does it occur only with ReLU neurons? Does it occur only with $L_2$ regularizers? Does it occur only in fully connected feedforward architectures? Does it occur only at the end of training? What happens if we iteratively balance neurons at random in a large network? And can partial or full balancing, before or during learning, be used as an effective regularization technique? All these questions, but the last one, are addressed by the theory of synaptic neural balance that we have developed and briefly describe in the next section. The last question, on using balancing as a learning regularizer, is the main topic of this paper and is addressed by the experiments presented in the following sections. Unless otherwise specified, throughout the paper, terms like "balancing" or "neural balancing" refer to "synaptic neural balancing".

## 2 The Theory of Synaptic Neural Balance

We present a brief summary of the main point of the theory. The complete theory is described in the Appendix with the detailed proofs of all the theorems. The first key point is that the base case described in the Introduction, can be extended in three main directions in terms of the activation functions, the regularizers, and the network architectures.

**Theorem:** (Balance and Regularizer Minimization) *Consider a neural network with BiLU activation functions in all the hidden units and overall error function of the form:*

$$\mathcal{E} = E(W) + R(W) \quad \text{with} \quad R(W) = \sum_w g_w(w) \qquad (2)$$

*where each function $g_w(w)$ is continuously differentiable, depends on the magnitude $|w|$ alone, and grows monotonically from $g_w(0) = 0$ to $g_w(+\infty) = +\infty$. For any setting of the weights $W$ and any hidden unit $i$ in the network and any $\lambda > 0$ we can multiply the incoming weights of $i$ by $\lambda$ and the*

*outgoing weights of $i$ by $1/\lambda$ without changing the overall error $E$. Then, for any neuron, there exists at least one optimal value $\lambda^*$ that minimizes $R(W)$. Any optimal value must be a solution of the consistency equation:*

$$\lambda^2 \sum_{w \in IN(i)} wg'_w(\lambda w) = \sum_{w \in OUT(i)} wg'_w(w/\lambda) \tag{3}$$

*Once the weights are rebalanced accordingly, the new weights must satisfy the generalized balance equation:*

$$\sum_{w \in IN(i)} wg'(w) = \sum_{w \in OUT(i)} wg'(w) \tag{4}$$

*In particular, if $g_w(w) = |w|^p$ for all the incoming and outgoing weights of neuron $i$, then the optimal value $\lambda^*$ is unique and equal to:*

$$\lambda^* = \Big( \frac{\sum_{w \in OUT(i)} |w|^p}{\sum_{w \in IN(i)} |w|^p} \Big)^{1/2p} = \Big( \frac{||OUT(i)||_p}{||IN(i)||_p} \Big)^{1/2} \tag{5}$$

*The decrease $\Delta R \geq 0$ in the value of the $L_p$ regularizer $R = \sum_w |w|^p$ is given by:*

$$\Delta R = \Big( \big( \sum_{w \in IN(i)} |w|^p \big)^{1/2} - \big( \sum_{w \in OUT(i)} |w|^p \big)^{1/2} \Big)^2 \tag{6}$$

*After balancing neuron $i$, its new weights satisfy the generalized $L_p$ balance equation:*

$$\sum_{w \in IN(i)} |w|^p = \sum_{w \in OUT(i)} |w|^p \tag{7}$$

*Proof:* The proof is given in the Appendix. We use the optimal value $\lambda^*$, which we proved how to find in the Appendix, for our experiments in the next Section.

*Network Architectures:* It is easy to see that the reasoning behind the base case can be applied to any BiLU neurons inside any architecture, such a fully connected feedforward, locally connected feedforward, or recurrent. Again this is because scaling does not change the effect of the neuron on the rest of the network and therefore we can always scale the neuron in a way that minimizes a particular cost function or regularizer. It is even possible to train a network with a certain regularizer and balance it with respect to a different regularizer. This brings us to the main result of the theory which is related to balancing algorithms. Imagine that we have a neural network containing BiLU (e.g. ReLU) neurons, with a fixed set of weights $W$. These could be the weights before learning has started, during learning (i.e. at a particular epoch), or after learning has finished. Imagine that we start balancing the BiLU neurons one after the other, in some regular order or, more generally, even in a stochastic order. Balancing the weights of a neuron may break the balance of another neuron. So while the value of the regularizer always decreases after each balancing operation, it is not clear what happens to the weights of the network, whether they converge to a stable value, and if so whether this value is unique. The main theorem of the theory is the proof that indeed not only the regularizer converges, but the weights themselves must converge and, most interestingly, they must converge to a unique point, which depends only on the initial set of weights $W$. The limit does *not* depend on the order in which the balancing operations are applied.

## 3  Related Work

Yang et al. [2022] proposed to replace the $L_2$ regularization term in the loss with the sum of products of l2 norms of the input and output weights. Stock et al. [2022] proposed a new local heterosynaptic learning rule by adding a kind of reconstruction loss term in which neurons try to balance themselves. Du et al. [2018] proved that gradient descent with infinitesimal step size effectively conserves the differences between squared norms of inputs and outputs weights of each layer without explicit regularization. Related results are also described in Arora et al. [2018]. Saul [2023] computes

| Type | No FB at Start | | | FB at Start | | |
|---|---|---|---|---|---|---|
| | Plain | L1 Reg. | L2 Reg. | Plain | L1 Reg. | L2 Reg. |
| 2 Layer FCN | 90.09% | 90.05% | 90.062% | **91.22%** | **93.96%** | **91.18%** |
| 3 Layer FCN | 89.594% | 89.67% | 89.70% | **90.83%** | **93.47%** | **90.79%** |
| 5 Layer FCN | 89.09% | 87.85% | 90.3% | **91.37%** | **95.50%** | **91.59%** |

Table 1: Test accuracy during training of Plain, L1 Regularized, and L2 Regularized Fully Connected Networks trained on MNIST, comparing full balancing before training with no full balance before training. Full balancing before training results in faster convergence, as well as universally higher attained test accuracy.

multiplicative rescaling factors—one at each hidden unit— to balance the weights of neural networks. Neyshabur et al. [2015a] shows that training with stochastic gradient descent does not work well in highly unbalanced neural networks, so they proposed a rescaling-invariant solution Neyshabur et al. [2015c]. Others have proposed that learning in neural networks can be accelerated with rescaling transformations Zhao et al. [2022], Armenta et al. [2023] without mentioning balancing the weights though. In our case we present both theoretical results on neural synaptic balance, including the existence and uniqueness of a globally balanced state (given an initial set of weights $W$), and experimental results showing that balancing neurons can expedite learning convergence and improve learning performance. This is also the first work experimenting with neural balance in both feedforward and recursive neural networks.

## 4    Experiments and Results

In our experiments, we train and compare various neural network architectures using full neural balancing, partial balancing, and $L_1$ or $L_2$ regularization. The term "plain" is used to refer to training of neural networks without balancing or regularizers. Full balance is obtained by iteratively balancing all BiLU neurons in the network until convergence is achieved. Partial balance is implemented by balancing the neurons in a layer-wise fashion, starting from the input layer and moving towards the output layer or vice-versa (no significant differences are observed). Due to the gradual nature of partial balance, the periodicity of the balancing operation is key to its implementation. In partial balance, the balancing operation can be performed up to once per epoch. Through the use of partial balancing during training, it has been observed that the ratio of the norms of a neuron's output to input weights tends to equalize, irrespective of the periodicity of epochs that we perform partial balancing operations. We have also observed that partial balancing helps the network converge faster and achieve a balanced state as is expected in a fully-trained network, same is in full balancing. The balancing operations for each neuron in each layer take place in parallel so they do not impose a bottleneck during training. Our results suggest that neural balancing is effective in training various types of neural networks with limited data. Furthermore, this approach proves beneficial in reducing overfitting and enhancing generalization in data-scarce environments.

To ensure reproducibility and fairness, experiments comparing training methodologies use the same range of seeds, learning rates, and train/test splits. Every experiment was run with 8 different seeds and the result reported is the average of them. A more detailed description of our experimental setup can be found in the Appendix. The roadmap of our experiments is organized as follows: first, we present experiments with the full dataset on both FCNs and RNNs. Then, we move onto data-scarce environments, amplifying the complexity of the experiments. For every experiment we deploy FCNs and RNNs ranging from smaller to larger sizes. The term FCN refers to Feedforward-layered networks with full connectivity between the layers.

### 4.1    Assessment of Full Balance Before Training

In table 1, we assess the use of the full balancing operation before the commencement of training. Compared to a standard initialization, the application of full balancing results in faster convergence, and higher overall accuracy when using the same model architecture, hyperparameters, and training methodologies. Partial balancing at every epoch after a full balance results in the least change due to the fundamentally similar nature of the full balancing operation to the partial balancing operation,

| Type | Plain | L2 NB | L1 NB | L2 1e-5 | L1 1e-5 |
|------|-------|-------|-------|---------|---------|
| 2-FCN | 91.22% | 91.19% | **94.542%** | 91.18% | 93.96% |
| 3-FCN | 90.84% | 90.86% | **93.94%** | 90.79% | 93.47% |
| 5-FCN | 91.37% | 91.63% | **96.26%** | 91.59% | 95.48% |

Table 2: Test accuracy across training comparisons of partial balancing, L2 Regularization, and Plain Accuracy for FCNs of varying sizes on MNIST. We observe that L1 partial balancing outperforms the other training methodologies on all model sizes

| Type | Plain | L1 Regularization | L2 Regularization |
|------|-------|-------------------|-------------------|
| No NB at Start | 88.26% | 88.23% | 88.22% |
| NB at Start | **88.64%** | 88.24% | 88.57% |

Table 3: Test accuracy for a Recurrent Neural Network trained on the IMDB sentiment analysis dataset, comparing Plain, L1 Regularized, and L2 Regularized models with and without a full balance at the start of training. A full balance before the commencement of training universally results in a higher test accuracy during training.

hence its omission from the plots. Repeated partial balancing results in the same outcome weights when using the same seed, albeit, over time since those weights aren't balanced from the start. Larger model sizes tend to exhibit a stronger correlation between the use of neural balancing, and the model's rate of convergence. These observations are especially exhibited in the normally trained models, where the use of full balancing at the start results in much faster convergence, as well as a higher final accuracy achieved by the model. table 1 displays the comparison between full balancing and no balancing performed on an FCN on MNIST before training. Each square in the grid represents a combination of a model size and a training methodology, where in every case, neural balancing results in an improvement in the rate of convergence and model accuracy.

## 4.2  Partial Balance with FCNs on MNIST

We test all forms of neural balancing on the MNIST handwritten digit dataset exclusively through FCNs. To fully capture the regularization capability of neural balancing, we test on a range of model architectures. From table 2, we observe that neural partial balancing results in faster convergence and test accuracy across all model sizes.

## 4.3  Full Balance with RNNs on IMBD

We continue our assessment of neural balancing with experiments performed on the RNN architecture. We train a 3-layered RNN on the IMDB sentiment analysis dataset, once again assessing full neural balancing with a 'plain', and regularized models. table 3 demonstrates that when full balancing is performed before training, the model has a better final accuracy when compared to equivalent, non-balanced methodlologies.

## 4.4  Neural Balance in Limited-data Environments

In data-scarce environments, models employing neural partial balancing techniques demonstrate accelerated convergence compared to unmodified models. These experiments are executed by stratifying samples equally according to their class labels to maintain a balanced distribution of classes within the training data. We observe that neural balancing results in higher accuracy and faster convergence, which could be attributed to better performance in data-scarce environments.

Further, we continue this study with an assessment of neural balancing on an RNN trained with a fraction of the original IMDB dataset. fig. 1 contains the comparison between training methodologies using a 3 layer RNN trained on the IMDB sentiment analysis dataset with 5% of the available training data. Partial balancing has higher accuracy on average, as well as a faster convergence, hinting at a characteristic of being able to generalize without a lot of training data.

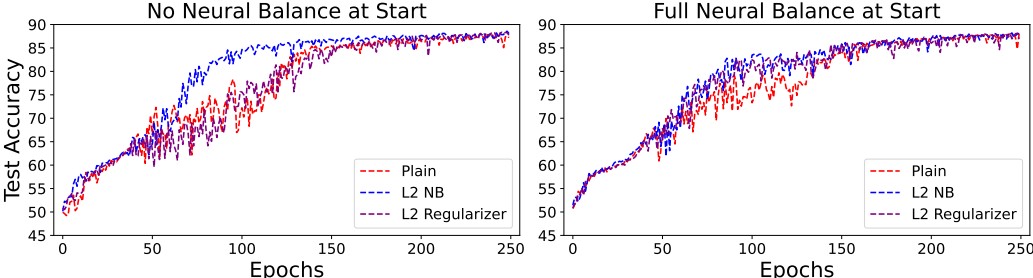

Figure 1: A comparison between partial balancing, Plain Accuracy, and L2 Regularization as performed on a 3 Layer RNN using 5% of the available dataset. Neural balancing reports the best overall performance.

| Type | No FB at Start | | | FB at Start | | |
|---|---|---|---|---|---|---|
| | 2 Layer FCN | 3 Layer FCN | 5 Layer FCN | 2 Layer FCN | 3 Layer FCN | 5 Layer FCN |
| Plain | 84.15% | 73.49% | 88.9% | **91.39%** | **91.42%** | **90.86%** |
| L1 NB | 91.25% | 90.57% | 92.87% | **93.26%** | **93.3%** | **92.92%** |
| L2 NB | 87.99% | 84.53% | 91.06% | **91.94%** | **91.37%** | **90.59%** |
| L1 Reg. | 83.92% | 72.03% | 11.35% | **85.99%** | **81.33%** | **19.79%** |
| L2 Reg. | 83.35% | 75.21% | 86.81% | **91.16%** | **90.86%** | **88.78%** |

Table 4: A comparison of test accuracy during training of various methodologies, using 1% of the MNIST dataset to simulate a limited data environment. The use of full balancing at the start of training not only increases the rate of convergence of every training methodology, but also allows for a higher attainable overall accuracy.

## 4.5   Discussion

Summing up our experiments we observe the following quantitative results. In FCNs, Neural Balance yields a notable improvement in model performance and convergence speed. Specifically, this method results in a 3-5% performance increase over plain models, and more than a 1% improvement over optimally L1-regularized models. Additionally, L1 neural balancing facilitates convergence at a rate 1.5 to 10 times faster, contingent on model size. When trained on limited datasets (1% of the full data), L1 neural balancing enhances performance by 3-10% compared to plain models, and by 1-5% relative to models regularized with L1 and L2 techniques. Moreover, it achieves up to a 10-fold increase in convergence speed, depending on model size. In RNNs, L1 neural balancing contributes to a 2-5% increase in convergence speed, with the application of L2 neural balancing leading to a more than 15% acceleration in convergence when training on 5% of the data. These findings underscore the efficacy of L1 neural balancing in optimizing both performance and training efficiency across different model architectures. We have extended our experiments due to page limits in the Appendix.

## 5   Conclusions

Synaptic balancing provides a novel approach to regularization that is supported by an underlying theory. Synaptic balancing is very general in the sense that it can be applied with all usual cost functions, including all $L_p$ cost functions. Synaptic balancing can be carried in full or in partial manner, due to the convexity connection provided by the main theorem. It can be applied at any time during the learning process: at the start of learning, at the end of learning, or during learning, by alternating balancing steps with stochastic gradient steps. Given, neural balance has some limitations; as mentioned earlier it can be applied only to neurons with specific activation functions (BiLU or slightly more general activation functions as shown in the Appendix). Another limitation is that it cannot be applied to neurons in Convolution layers due to the nature of the convolution operation with the kernels. Simulations show that these approaches can improve learning in terms of speed (fewer epochs), accuracy or generalization abilities. Thus, in short, balancing is a novel effective approach to regularization that can be added to the list of tools available to regularize networks, like dropout, and other regularization tools. Finally, a neuron can balance its weights independently of all

other neurons in the network. The knowledge required for balancing is entirely *local* and available at each neuron. In this sense, balancing is a natural algorithm for distributed asynchronous architectures and physical neural systems, and as such it may find applications in neuromorphic chip designs or brain studies.

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

# A    Appendix / supplemental material

# B    Appendix

Here we detail the additional theory, datasets, models, and training procedures used in the experiments in the main paper, separated into subsections which correspond to that of the main paper. We also included some supplemental experiments that are not present in the main paper.

In order to ensure that our results are reproducible, when we compare training methodologies, we do so using a sample size of 8 different, and random, seeds per methodology, with those seeds being shared with the other training methodologies. We train all of our models on a server equipped with 8 Nvidia RTX A6000 Ada Generation graphics cards, with 384 GB of total memory, run on CUDA version 12.4.

## B.1    Establishing Partial Balancing

In our experiments, we annotate 2 different kinds of neural balancing operations: L1 Neural Balancing, and L2 Neural Balancing. The names represent the norms used when balancing the input and output weights, with the L1 norm being used for L1 Neural Balancing, and the L2 norm being used for L2 Neural Balancing.

## B.2    Toy Experiment on a Circle Toy Dataset

To validate our initial hypothesis, which is that the balancing operation results in the equalization of the norms of the input and output weights for every neuron in a neural network, we observe the ratio between the aforementioned norms during training. We do this through a toy network trained on a simple 2-dimensional dataset for a binary classification task, where the limited number of layers and 'neurons' allow us to measure weights without the computational intensity attributed to accessing values from a large network. We compare the use of full balancing with partial balancing during training. Both methodologies result in the optimal factor $\lambda^*$ calculated during balancing to converge to 1, confirming that the norms of the input and output weights for each neuron equalize through the use of balancing. fig. 2 contains partial balancing performed every epoch on a 5-neuron toy model trained on a 2-dimensional concentric circle toy dataset showing that the input and output weight norms equalize for each neuron.

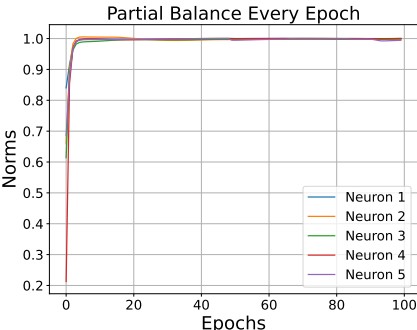

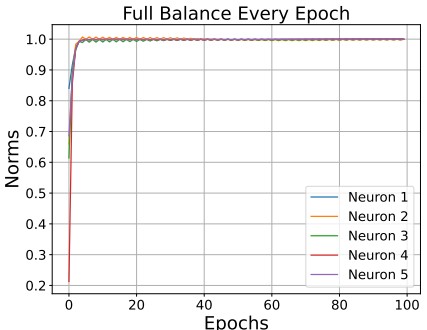

Figure 2: Partial balancing performed every epoch on a 5-neuron toy model trained on a 2-dimensional dataset for a binary classification task showing that the input and output weight norms equalize for each neuron

Figure 3: Full balancing performed every epoch on a 5-neuron toy model trained on a 2-dimensional dataset for a binary classification task showing that the input and output weight norms equalize for each neuron

To contextualize the rate of convergence of the norms from the partial balancing toy experiment, we measure the input and output norms of each neuron after a full-balance has been performed on the network. While the full-balance guarantees that the input and output norms of each neuron will always be close to each other, since full balancing is performed until that requirement is met, it remains useful as a benchmark for the rate of convergence of partial-balancing. fig. 3 delineates

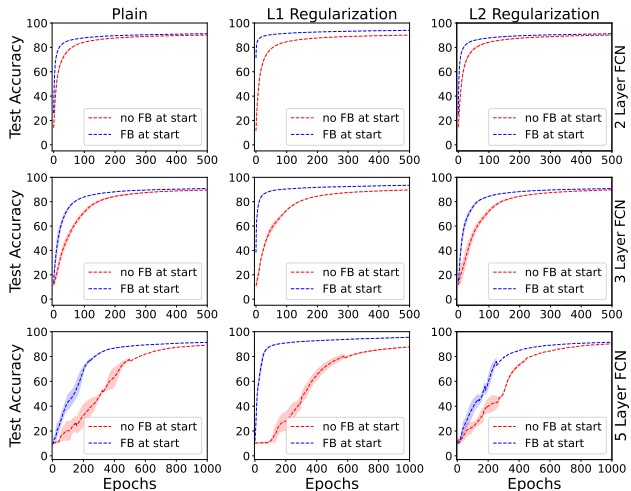

Figure 4: A demonstration of the effect of a full neural balance before the start of training on various sizes of fully connected networks, using various training methodologies. Regardless of L2 Regularization, neural partial balancing, or plain accuracy used in training, a neural full balance results in faster convergence, and a higher overall accuracy.

| Type | No FB at Start | | | FB at Start | | |
|---|---|---|---|---|---|---|
| | Plain | L1 Reg. | L2 Reg. | Plain | L1 Reg. | L2 Reg. |
| 2 Layer FCN | 90.09% | 90.05% | 90.062% | **91.22%** | **93.96%** | **91.18%** |
| 3 Layer FCN | 89.594% | 89.67% | 89.70% | **90.83%** | **93.47%** | **90.79%** |
| 5 Layer FCN | 89.09% | 87.85% | 90.3% | **91.37%** | **95.50%** | **91.59%** |

Figure 5: Accompanying fig. 4, Test accuracy during training of Plain, L1 Regularized, and L2 Regularized Fully Connected Networks trained on MNIST, comparing full balancing before training with no full balance before training. As observed in fig. 4, full balancing before training results in faster convergence, as well as universally higher attained test accuracy.

the rate of convergence of the input and output norms, doing so almost immediately, due to the methodology of full balancing. fig. 2 demonstrates the efficacy of partial-balancing, resulting in a rapid, and computationally less expensive method of 'balancing' neurons.

## B.3    Assessment of Full Balance Before Training

In the main paper, we assess the use of the full balancing operation before the start of training to demonstrate its efficacy at increasing the rate of convergence and overall test accuracy of various model architectures and training styles. Partial balancing at every epoch after a full balance results in the least change due to the fundamentally similar nature of the full balancing operation to the partial balancing operation, hence its omission from the plots. Repeated partial balancing results in wthe same outcome weights when using the same seed, albeit, over time since those weights aren't balanced from the start. In these experiments, we use fully connected neural networks in a few sizes to demonstrate the range of the balancing operation. Full balance before training is shown to increase the rate of convergence, as well as the overall accuracy obtainable during training. To assess full neural balance before training, we performed a full balancing operation on the neurons of the model after the initialization of the model's weights, and before the commencement of training.

## B.4    Partial Balance with FCNs

In the main paper, we assess the use of the partial balancing operation during training to demonstrate its efficacy at increasing the rate of convergence and overall test accuracy of various model architectures and training styles. As included in the main paper in section 4.2, we supplement our tabular results in fig. 6 with plots that delineate the positive impact of partial and full neural balance as performed through the balancing operation during/before training. Following the line of inquiry

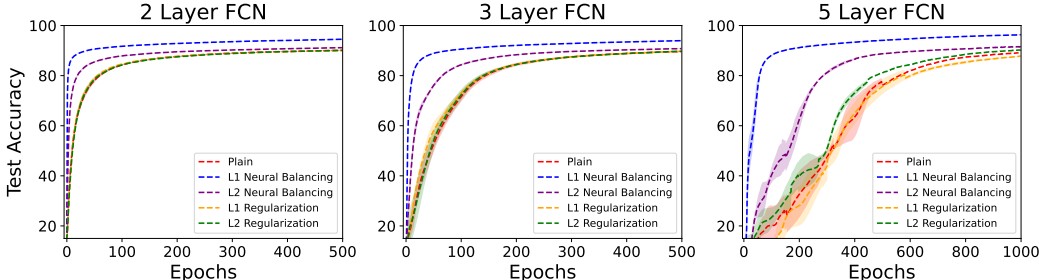

Figure 6: Accompanying table 2, comparison of neural balance, L1 and L2 Regularization on MNIST. We observe that as the models grow bigger, neural balance helps model converge faster and perform better than the other techniques.

| Type | Plain | L2 NB | L1 NB | L2 1e-5 | L1 1e-5 |
|------|-------|-------|-------|---------|---------|
| 2-FCN | 91.22% | 91.19% | **94.542%** | 91.18% | 93.96% |
| 3-FCN | 90.84% | 90.86% | **93.94%** | 90.79% | 93.47% |
| 5-FCN | 91.37% | 91.63% | **96.26%** | 91.59% | 95.48% |

Table 5: Test accuracy across training comparisons of partial balancing, L2 Regularization, and Plain Accuracy for FCNs of varying sizes on MNIST. We observe that L1 partial balancing outperforms the other training methodologies on all model sizes

on the performance of neural balancing on FCNs trained on MNIST, we assess its performance on FashionMNIST using the same model architectures. We use FCNs of various sizes, and perform a partial balance on the model at every epoch, identically to the MNIST experiments. We observed similar results on performance and convergence on FashionMNIST. Regardless of the size of the model, or the methodology used to train said model, neural balancing significantly increases the rate of convergence, as well as its overall test accuracy.

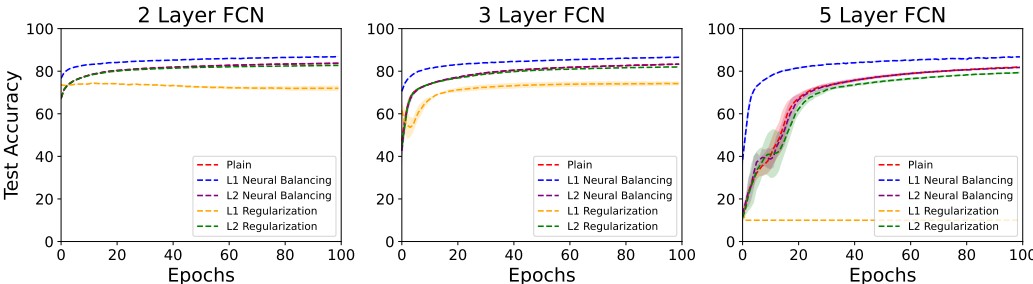

Figure 7: Test accuracy across training comparisons of partial balancing, L2 Regularization, and Plain Accuracy for FCNs of varying sizes on Fashion MNIST. We observe that L1 partial balancing outperforms the other training methodologies on all model sizes.

## B.5   Full Balance with RNNs on IMDB

In the main paper, we assess the use of the partial balancing operation during training to demonstrate its efficacy at increasing the rate of convergence and overall test accuracy of a recurrent neural network architecture, comparing various training styles in the process. For these experiments, we use the IMDB sentiment analysis dataset. The IMDB dataset is a collection of positively/negatively labeled text containing movie reviews from the popular movie review website IMDB. We use a recurrent neural network with 3 hidden layers to demonstrate the efficacy of the partial balancing operation.

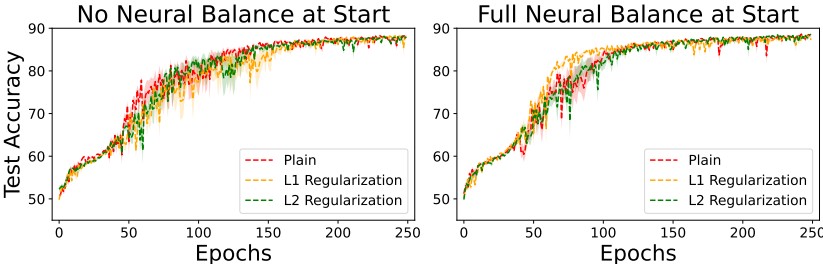

Figure 8: A comparison between partial balancing, L2 Regularization, and Plain Accuracy on a 3 Layer RNN using the IMDB sentiment analysis dataset. We also contrast the standard initialization with a full neural balancing operation performed before the start of training. We observe that neural partial balancing performed every epoch, paired with a full balance before training, results in the best overall accuracy, and convergence speed.

| Type | Plain | L1 Regularization | L2 Regularization |
|------|-------|-------------------|-------------------|
| No NB at Start | 88.26 | 88.23 | 88.22 |
| NB at Start | **88.64%** | **88.24%** | **88.57%** |

Table 6: Accompanying fig. 8, Test accuracy for a Recurrent Neural Network trained on the IMDB sentiment analysis dataset, comparing Plain, L1 Regularized, and L2 Regularized models with and without a full balance at the start of training. A full balance before the commencement of training universally results in a higher test accuracy during training.

## B.6 Neural Balance in Limited Data Environments

As mentioned in the main paper, we assess the performance of a full neural balance, as well as partial balance during training. These experiments are executed by stratifying samples equally according to their class labels to maintain a balanced distribution of classes within the training data. Accompanying table 4, we add plots to visualize the tabular information, and to demonstrate the efficacy of neural balance at incresing the rate of convergence of training. fig. 9 delineates the efficacy of partial balance at improving overall accuracy and training speed.

## B.7 Neural Balancing in Transformers

Transformers models, characterized by their attention mechanism, represent the state of the art in the field of Natural Language Processing. In our study, neural balancing is only applied to the feed-forward, linear layers in the transformer block, as any manipulation of the attention matrix strongly affects the model output. We observe that the best training method is the 'clean' style, where neither neural balancing, nor L2 regularization is applied to the model. For these experiments, we use

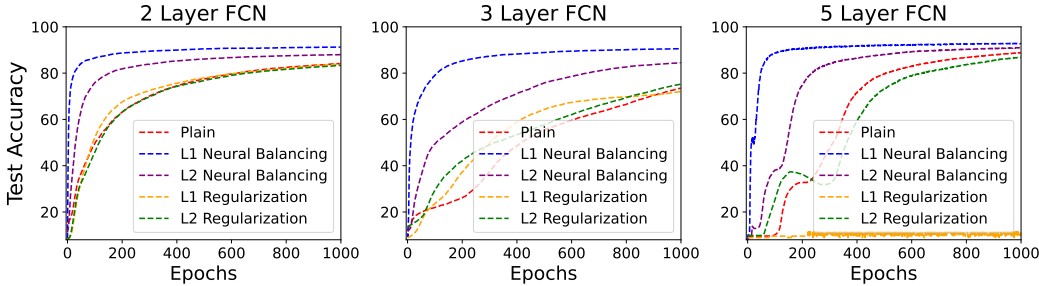

Figure 9: A comparison between partial balance, standard regularization, and Plain Accuracy, on various Fully Connected Networks trained on 1% of the MNIST dataset. We observe that neural balancing consistently has a positive impact on the rate of convergence and overall accuracy of the model.

| Type | Plain | L1 Regularization | L2 Regularization |
|---|---|---|---|
| No NB at Start | **83.66%** | 81.95% | 83.36% |
| NB at Start | 83.52% | 81.65% | 83.21% |

Table 7: Accompanying fig. 10, Test accuracy for a Transformer Network trained on the IMDB sentiment analysis dataset, comparing Plain, L1 Regularized, and L2 Regularized models with and without a full balance at the start of training.

the IMDB sentiment analysis dataset, and we use a transformer model with 8 attention heads, and 6 feedforward encoder layers, each with a hidden dimensionality of 2048 units.

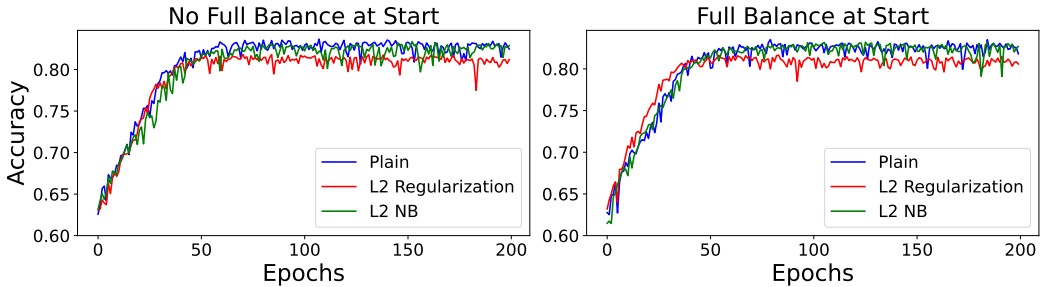

Figure 10: A comparison of various combinations of full balancing and training methodologies using a transformer model. The combination of L2 regularization and neural balancing fails after some epochs, and the clean model without any form of balancing performs the best out of the training styles.

### B.8 Neural Balance in Bioplausible Architectures

In the main paper, we detail the use of neural balancing operations in biologically plausible systems. Specifically, we employ Direct Feedback Alignment (DFA) in place of backpropagation as the biologically plausible alternative, and perform partial balancing during the training of the model to achieve neural balance.

| Type | Accuracy |
|---|---|
| clean | 97.764% |
| nb | 97.764% |
| L2 with $\lambda = 1e-4$ | 97.758% |
| L2 with $\lambda = 1e-5$ | 97.764% |

Figure 11: Comparison between neural balancing and L2 with various lambda values using a 'clean' model as a benchmark, trained with DFA on a 2-layer fully connected network

| Type | Accuracy |
|---|---|
| clean | 97.4525% |
| nb | 97.4525% |
| L2 with $\lambda = 1e-4$ | 95.417% |
| L2 with $\lambda = 1e-5$ | 97.4525 |

Figure 12: Comparison between neural balancing and L2 with various lambda values using a 'clean' model as a benchmark, trained with DFA on a 7-layer fully connected network

## C   Full Proof and Theory

### C.1   Homogeneous and BiLU Activation Functions

In this section, we generalize the basic example of the introduction from the standpoint of the activation functions. In particular, we consider homogeneous activation functions (defined below). The importance of homogeneity has been previously identified in somewhat different contexts Neyshabur et al. [2015b]. Intuitively, homogeneity is a form of linearity with respect to weight scaling and thus it is useful to motivate the concept of homogeneous activation functions by looking at other notions of linearity for activation functions. This will also be useful for Section C.5 where even more general classes of activation functions are considered.

### C.1.1 Additive Activation Functions

**Definition C.1.** *A neuronal activation function $f : \mathbb{R} \to \mathbb{R}$ is additively linear if and only if $f(x + y) = f(x) + (f(y)$ for any real numbers $x$ and $y$.*

**Proposition C.2.** *The class of additively linear activation functions is exactly equal to the class of linear activation functions, i.e., activation functions of the form $f(x) = ax$.*

*Proof.* Obviously linear activation functions are additively linear. Conversely, if $f$ is additively linear, the following three properties are true:

(1) One must have: $f(nx) = nf(x)$ and $f(x/n) = f(x)/n$ for any $x \in \mathbb{R}$ and any $n \in \mathbb{N}$. As a result, $f(n/m) = nf(1)/m$ for any integers $n$ and $m$ ($m \neq 0$).

(2) Furthermore, $f(0 + 0) = f(0) + f(0)$ which implies: $f(0) = 0$.

(3) And thus $f(x - x) = f(x) + f(-x) = 0$, which in turn implies that $f(-x) = -f(x)$.

From these properties, it is easy to see that $f$ must be continuous, with $f(x) = xf(1)$, and thus $f$ must be linear. $\square$

### C.1.2 Multiplicative Activation Functions

**Definition C.3.** *A neuronal activation function $f : \mathbb{R} \to \mathbb{R}$ is multiplicative if and only if $f(xy) = f(x)(f(y)$ for any real numbers $x$ and $y$.*

**Proposition C.4.** *The class of continuous multiplicative activation functions is exactly equal to the class of functions comprising the functions: $f(x) = 0$ for every $x$, $f(x) = 1$ for every $x$, and all the even and odd functions satisfying $f(x) = x^c$ for $x \geq 0$, where $c$ is any constant in $\mathbb{R}$.*

*Proof.* It is easy to check the functions described in the proposition are multiplicative. Conversely, assume $f$ is multiplicative. For both $x = 0$ and $x = 1$, we must have $f(x) = f(xx) = f(x)f(x)$ and thus $f(0)$ is either 0 or 1, and similarly for $f(1)$. If $f(1) = 0$, then for any $x$ we must have $f(x) = 0$ because: $f(x) = f(1x) = f(1)f(x) = 0$. Likewise, if $f(0) = 1$, then for any $x$ we must have $f(x) = 1$ because: $1 = f(0) = f(0x) = f(0)f(x) = f(x)$. Thus, in the rest of the proof, we can assume that $f(0) = 0$ and $f(1) = 1$. By induction, it is easy to see that for any $x \geq 0$ we must have: $f(x^n) = f(x)^n$ and $f(x^{1/n}) = (f(x))^{1/n}$ for any integer (positive or negative). As a result, for any $x \in \mathbb{R}$ and any integers $n$ and $m$ we must have: $f(x^{n/m}) = f(x)^{n/m}$. By continuity this implies that for any $x \geq 0$ and any $r \in R$, we must have: $f(x^r) = f(x)^r$. Now there is some constant $c$ such that: $f(e) = e^c$. And thus, for any $x > 0$, $f(x) = f(e^{\log x}) = [f(e)]^{\log x} = e^{c \log x} = x^c$. To address negative values of $x$, note that we must have $f[(-1)(-1 = f(1) = 1f(-1)^2$. Thus, $f(-1)$ is either equal to 1 or to -1. Since for any $x > 0$ we have $f(-x) = f(-1)f(x)$, we see that if $f(-1) = 1$ the function must be even ($f(-x) = f(x) = x^c$), and if $f(-1) = -1$ the function must be odd ($f(-x) = -f(x)$). $\square$

We will return to multiplicative activation function in a later section.

### C.1.3 Linearly Scalable Activation Functions

**Definition C.5.** *A neuronal activation function $f : \mathbb{R} \to \mathbb{R}$ is linearly scalable if and only if $f(\lambda x) = \lambda f(x)$ for every $\lambda \in \mathbb{R}$.*

**Proposition C.6.** *The class of linearly scalable activation functions is exactly equal to the class of linear activation functions, i.e., activation functions of the form $f(x) = ax$.*

*Proof.* Obviously, linear activation functions are linearly scalable. For the converse, if $f$ is linearly multiplicative we must have $f(\lambda x) = \lambda f(x) = xf(\lambda)$ for any $x$ and any $\lambda$. By taking $\lambda = 1$, we get $f(x) = f(1)x$ and thus $f$ is linear. $\square$

Thus the concepts of linearly additive or linearly scalable activation function are of limited interest since both of them are equivalent to the concept of linear activation function. A more interesting class is obtained if we consider linearly scalable activation functions, where the scaling factor $\lambda$ is constrained to be positive ($\lambda > 0$), also called homogeneous functions.

### C.1.4 Homogeneous Activation Functions

**Definition C.7.** *(Homogeneous) A neuronal activation function $f : \mathbb{R} \to \mathbb{R}$ is homogeneous if and only if: $f(\lambda x) = \lambda f(x)$ for every $\lambda \in \mathbb{R}$ with $\lambda > 0$.*

**Remark C.8.** *Note that if $f$ is homogeneous, $f(\lambda 0) = \lambda f(0) = f(0)$ for any $\lambda > 0$ and thus $f(0) = 0$. Thus it makes no difference in the definition of homogeneous if we set $\lambda \geq 0$ instead of $\lambda > 0$).*

**Remark C.9.** *Clearly, linear activation functions are homogeneous. However, there exists also homogeneous functions that are non-linear, such as ReLU or leaky ReLU activation functions.*

We now provide a full characterization of the class of homogeneous activation functions.

### C.1.5 BiLU Activation Functions

We first define a new class of activation functions, corresponding to bilinear units (BiLU), consisting of two half-lines meeting at the origin. This class contains all the linear functions, as well as the ReLU and leaky ReLU functions, and many other functions.

**Definition C.10.** *(BiLU) A neuronal activation function $f : \mathbb{R} \to \mathbb{R}$ is bilinear (BiLU) if and only if $f(x) = ax$ when $x < 0$, and $f(x) = bx$ when $x \geq 0$, for some fixed parameters $a$ and $b$ in $\mathbb{R}$.*

These include linear units ($a = b$), ReLU units ($a = 0, b = 1$), leaky ReLU ($a = \epsilon; b = 1$) units, and symmetric linear units ($a = -b$), all of which can also be viewed as special cases of piece-wise linear units Tavakoli et al. [2021], with a single hinge. One advantage of ReLU and more generally BiLU neurons, which is very important during backpropagation learning, is that their derivative is very simple and can only take one of two values ($a$ or $b$).

**Proposition C.11.** *A neuronal activation function $f : \mathbb{R} \to \mathbb{R}$ is homogeneous if and only if it is a BiLU activation function.*

*Proof.* Every function in BiLU is clearly homogeneous. Conversely, any homogeneous function $f$ must satisfy: (1) $f(0x) = 0f(x) = f(0) = 0$; (2)$f(x) = f(1x) = f(1)x$ for any positive $x$; and (3) $f(x) = f(-u) = f(-1)u = -f(-1)x$ for any negative $x$. Thus $f$ is in BiLU with $a = -f(-1)$ and $b = f(1)$. $\qquad\square$

In Appendix A, we provide a simple proof that networks of BiLU neurons, even with a single hidden layer, have universal approximation properties. In the next two sections, we introduce two fundamental neuronal operations, scaling and balancing, that can be applied to the incoming and outgoing synaptic weights of neurons with BiLU activation functions.

### C.2 Scaling

**Definition C.12.** *(Scaling) For any BiLU neuron $i$ in network and any $\lambda > 0$, we let $S_\lambda(i)$ denote the synaptic scaling operation by which the incoming connection weights of neuron $i$ are multiplied by $\lambda$ and the outgoing connection weights of neuron $i$ are divided by $\lambda$.*

Note that because of the homogeneous property the scaling operation does not change how neuron $i$ affects the rest of the network. In particular, the input-output function of the overall network remains unchanged after scaling neuron $i$ bt any $\lambda > 0$. Note also that scaling always preserves the sign of the synaptic weights to which it is applied, and the scaling operation can never convert a non-zero synaptic weight into a zero synaptic weight, or vice versa.

As usual, the bias is treated here as an additional synaptic weight emanating from a unit clamped to the value one. Thus scaling is applied to the bias.

**Proposition C.13.** *(Commutativity of Scaling) Scaling operations applied to any pair of BiLU neurons $i$ and $j$ in a neural network commute: $S_\lambda(i)S_\mu(j) = S_\mu(j)S_\lambda(i)$, in the sense that the resulting network weights are the same, regardless of the order in which the scaling operations are applied. Furthermore, for any BiLU neuron $i$: $S_\lambda(i)S_\mu(i) = S_\mu(i)S_\lambda(i) = S_{\lambda\mu}(i)$.*

This is obvious. As a result, any set $I$ of BiLU neurons in a network can be scaled simultaneously or in any sequential order while leading to the same final configuration of synaptic weights. If we denote

by $1, 2, \ldots, n$ the neurons in $I$, we can for instance write: $\prod_{i \in I} S_{\lambda_i}(i) = \prod_{\sigma(i) \in I} S_{\lambda_{\sigma(i)}}(\sigma(i))$ for any permutation $\sigma$ of the neurons. Likewise, we can collapse operations applied to the same neuron. For instance, we can write: $S_5(1)S_2(2)S_3(1)S_4(2) = S_{15}(1)S_8(2) = S_8(2)S_{15}(1)$

**Definition C.14.** *(Coordinated Scaling) For any set $I$ of BiLU neurons in a network and any $\lambda > 0$, we let $S_\lambda(I)$ denote the synaptic scaling operation by which all the neurons in $I$ are scaled by the same $\lambda$.*

## C.3   Balancing

**Definition C.15.** *(Balancing) Given a BiLU neuron in a network, the balancing operation $B(i)$ is a particular scaling operation $B(i) = S_{\lambda^*}(i)$, where the scaling factor $\lambda^*$ is chosen to optimize a particular cost function, or regularizer, asociated with the incoming and outgoing weights of neuron $i$.*

For now, we can imagine that this cost function is the usual $L_2$ (least squares) regularizer, but in the next section, we will consider more general classes of regularizers and study the corresponding optimization process. For the $L_2$ regularizer, as shown in the next section, this optimization process results in a unique value of $\lambda^*$ such that sum of the squares of the incoming weights is equal to the sum of the squares of the outgoing weights, hence the term "balance". Note that obviously $B(B(i)) = B(i)$ and that, as a special case of scaling operation, the balancing operation does not change how neuron $i$ contributes to the rest of the network, and thus it leaves the overall input-output function of the network unchanged.

Unlike scaling operations, balancing operations in general do not commute as balancing operations (they still commute as scaling operations). Thus, in general, $B(i)B(j) \neq B(j)B(i)$. This is because if neuron $i$ is connected to neuron $j$, balancing $i$ will change the connection between $i$ and $j$, and, in turn, this will change the value of the optimal scaling constant for neuron $j$ and vice versa. However, if there are no non-zero connections between neuron $i$ and neuron $j$ then the balancing operations commute since each balancing operation will modify a different, non-overlapping, set of weights.

**Definition C.16.** *(Disjoint neurons) Two neurons $i$ and $j$ in a neural network are said to be disjoint if there are no non-zero connections between $i$ and $j$.*

Thus in this case $B(i)B(j) = S_{\lambda^*}(i)S_{\mu^*}(j) = S_{\mu^*}(j)S_{\lambda^*}(i) = B(j)B(i)$. This can be extended to disjoint sets of neurons.

**Definition C.17.** *(Disjoint Set of Neurons) A set $I$ of neurons is said to be disjoint if for any pair $i$ and $j$ of neurons in $I$ there are no non-zero connections between $i$ and $j$.*

For example, in a layered feedforward network, all the neurons in a layer form a disjoint set, as long as there are no intra-layer connections or, more precisely, no non-zero intra-layer connections. All the neurons in a disjoint set can be balanced in any order resulting in the same final set of synaptic weights. Thus we have:

**Proposition C.18.** *If we index by $1, 2, \ldots, n$ the neurons in a disjoint set $I$ of BiLU neurons in a network, we have: $\prod_{i \in I} B(i) = \prod_{i \in I} S_{\lambda_i^*}(i) = \prod_{\sigma(i) \in I} S_{\lambda_{\sigma(i)}^*}(\sigma(i)) = \prod_{\sigma(i) \in I} B(\sigma(i))$ for any permutation $\sigma$ of the neurons.*

Finally, we can define the coordinated balancing of any set $I$ of BiLU neurons (disjoint or not disjoint).

**Definition C.19.** *(Coordinated Balancing) Given any set $I$ of BiLU neurons (disjoint or not disjoint) in a network, the coordinated balacing of these neurons, written as $B_{\lambda^*}(I)$, corresponds to coordinated scaling all the neurons in $I$ by the same factor $\lambda^*$, Where $\lambda^*$ minimizes the cost functions of all the weights, incoming and outgoing, associated with all the neurons in $I$.*

**Remark C.20.** *While balancing corresponds to a full optimization of the scaling operation, it is also possible to carry a partial optimization of the scaling operation by choosing a scaling factor that reduces the corresponding contribution to the regularizer without minimizing it.*

## C.4   General Framework and Single Neuron Balance

In this section, we generalize the kinds of regularizer to which the notion of neuronal synaptic balance can be applied, beyond the usual $L_2$ regularizer and derive the corresponding balance equations.

Thus we consider a network (feedforward or recurrent) where the hidden units are BiLU units. The visible units can be partitioned into input units and output units. For any hidden unit $i$, if we multiply all its incoming weights $IN(i)$ by some $\lambda > 0$ and all its outgoing weights $OUT(i)$ by $1/\lambda$ the overall function computed by the network remains unchanged due to the BiLU homogeneity property. In particular, if there is an error function that depends uniquely on the input-output function being computed, this error remains unchanged by the introduction of the multiplier $\lambda$. However, if there is also a regularizer $R$ for the weights, its value is affected by $\lambda$ and one can ask what is the optimal value of $\lambda$ with respect to the regularizer, and what are the properties of the resulting weights. This approach can be applied to any regularizer. For most practical purposes, we can assume that the regularizer is continuous in the weights (hence in $\lambda$) and lower-bounded. Without any loss of generality, we can assume that it is lower-bounded by zero. If we want the minimum value to be achieved by some $\lambda > 0$, we need to add some mild condition that prevents the minimal value to be approached as $\lambda \to 0)$, or as $\lambda \to +\infty$. For instance, it is enough if there is an interval $[a, b]$ with $0 < a < b$ where $R$ achieves a minimal value $R_{min}$ and $R \geq R_{min}$ in the intervals $(0, a]$ and $[b, +\infty)$. Additional (mild) conditions must be imposed if one wants the optimal value of $\lambda$ to be unique, or computable in closed form (see Theorems below). Finally, we want to be able to apply the balancing approach

Thus, we consider overall regularized error functions, where the regularizer is very general, as long as it has an additive form with respect to the individual weights:

$$\mathcal{E}(W) = E(W) + R(W) \quad \text{with} \quad R(W) = \sum_w g_w(w) \tag{8}$$

where $W$ denotes all the weights in the network and $E(W)$ is typically the negative log-likelihood (LMS error in regression tasks, or cross-entropy error in classification tasks). We assume that the $g_w$ are continuous, and lower-bounded by 0. To ensure the existence and uniqueness of minimum during the balancing of any neuron, We will assume that each function $g_w$ depends only on the magnitude $|w|$ of the corresponding weight, and that $g_w$ is monotonically increasing from 0 to $+\infty$ ($g_w(0) = 0$ and $\lim_{x \to +\infty} g_w(x) = +\infty$). Clearly, $L_2, L_1$ and more generally all $L_p$ regularizers are special cases where, for $p > 0$, $L^p$ regularization is defined by: $R(W) = \sum_w |w|^p$.

When indicated, we may require also that the functions $g_w$ be continuously differentiable, except perhaps at the origin in order to be able to differentiate the regularizer with respect to the $\lambda$'s and derive closed form conditions for the corresponding optima. This is satisfied by all forms of $L_p$ regularization, for $p > 0$.

**Remark C.21.** *Often one introduces scalar multiplicative hyperparameters to balance the effect of $E$ and $R$, for instance in the form: $\mathcal{E} = E + \beta R$. These cases are included in the framework above: multipliers like $\beta$ can easily be absorbed into the functions $g_w$ above.*

**Theorem C.22.** *(General Balance Equation). Consider a neural network with BiLU activation functions in all the hidden units and overall error function of the form:*

$$\mathcal{E} = E(W) + R(W) \quad \text{with} \quad R(W) = \sum_w g_w(w) \tag{9}$$

*where each function $g_w(w)$ is continuous, depends on the magnitude $|w|$ alone, and grows monotonically from $g_w(0) = 0$ to $g_w(+\infty) = +\infty$. For any setting of the weights $W$ and any hidden unit $i$ in the network and any $\lambda > 0$ we can multiply the incoming weights of $i$ by $\lambda$ and the outgoing weights of $i$ by $1/\lambda$ without changing the overall error $E$. Furthermore, there exists a unique value $\lambda^*$ where the corresponding weights $v$ ($v = \lambda^* w$ for incoming weights, $v = w/\lambda^*$ for the outgoing weights) achieve the balance equation:*

$$\sum_{v \in IN(i)} g_w(v) = \sum_{w \in OUT(i)} g_w(v) \tag{10}$$

*Proof.* Under the assumptions of the theorem, $E$ is unchanged under the rescaling of the incoming and outgoing weights of unit $i$ due to the homogeneity property of BiLUs. Without any loss of generality, let us assume that at the beginning: $\sum_{w \in IN(i)} g_w(w) < \sum_{w \in OUT(i)} g_w(w)$. As we increase $\lambda$ from 1 to $+\infty$, by the assumptions on the functions $g_w$, the term $\sum_{w \in IN(i)} g_w(\lambda w)$ increases continuously

from its initial value to $+\infty$, whereas the term $\sum_{w\in OUT(i)} g_w)w/\lambda)$ decreases continuously from its initial value to 0. Thus, there is a unique value $\lambda^*$ where the balance is realized. If at the beginning $\sum_{w\in IN(i)} g_w(w) > \sum_{w\in OUT(i)} g_w(w)$, then the same argument is applied by decreasing $\lambda$ from 1 to 0. $\qquad\square$

**Remark C.23.** *For simplicity, here and in other sections, we state the results in terms of a network of BiLU units. However, the same principles can be applied to networks where only a subset of neurons are in the BiLU class, simply by applying scaling and balancing operations to only those neurons. Furthermore, not all BiLU neurons need to have the same BiLU activation functios. For instance, the results still hold for a mixed network containing both ReLU and linear units.*

**Remark C.24.** *In the setting of Theorem C.22, the balance equations do not necessarily minimize the corresponding regularization term. This is addressed in the next theorem.*

**Remark C.25.** *Finally, zero weights ($w = 0$) can be ignored entirely as they play no role in scaling or balancing. Furthermore, if all the incoming or outgoing weights of a hidden unit were to be zero, it could be removed entirely from the network*

**Theorem C.26.** *(Balance and Regularizer Minimization) We now consider the same setting as in Theorem C.22, but in addition we assume that the functions $g_w$ are continuously differentiable, except perhaps at the origin. Then, for any neuron, there exists at least one optimal value $\lambda^*$ that minimizes $R(W)$. Any optimal value must be a solution of the consistency equation:*

$$\lambda^2 \sum_{w\in IN(i)} wg_w'(\lambda w) = \sum_{w\in OUT(i)} wg_w'(w/\lambda) \tag{11}$$

*Once the weights are rebalanced accordingly, the new weights must satisfy the generalized balance equation:*

$$\sum_{w\in IN(i)} wg'(w) = \sum_{w\in OUT(i)} wg'(w) \tag{12}$$

*In particular, if $g_w(w) = |w|^p$ for all the incoming and outgoing weights of neuron $i$, then the optimal value $\lambda^*$ is unique and equal to:*

$$\lambda^* = \Big(\frac{\sum_{w\in OUT(i)} |w|^p}{\sum_{w\in IN(i)} |w|^p}\Big)^{1/2p} = \Big(\frac{||OUT(i)||_p}{||IN(i)||_p}\Big)^{1/2} \tag{13}$$

*The decrease $\Delta R \geq 0$ in the value of the $L_p$ regularizer $R = \sum_w |w|^p$ is given by:*

$$\Delta R = \Big(\big(\sum_{w\in IN(i)} |w|^p\big)^{1/2} - \big(\sum_{w\in OUT(i)} |w|^p\big)^{1/2}\Big)^2 \tag{14}$$

*After balancing neuron $i$, its new weights satisfy the generalized $L_p$ balance equation:*

$$\sum_{w\in IN(i)} |w|^p = \sum_{w\in OUT(i)} |w|^p \tag{15}$$

*Proof.* Due to the additivity of the regularizer, the only component of the regularizer that depends on $\lambda$ has the form:

$$R(\lambda) = \sum_{w\in IN(i)} g_w(\lambda w) + \sum_{w\in OUT(i)} g_w(w/\lambda) \tag{16}$$

Because of the properties of the functions $g_w$, $R_\lambda$ is continously differentiable and strictly bounded below by 0. So it must have a minimum, as a function of $\lambda$ where its derivative is zero. Its derivative with respect to $\lambda$ has the form:

$$R'(\lambda) = \sum_{w\in IN(i)} wg_w'(\lambda w) + \sum_{w\in OUT(i)} (-w/\lambda^2)g_w'(w/\lambda) \tag{17}$$

Setting the derivative to zero, gives:

$$\lambda^2 \sum_{w \in IN(i)} w g'_w(\lambda w) = \sum_{w \in OUT(i)} w g'_w(w/\lambda) \tag{18}$$

Assuming that the left-hand side is non-zero, which is generally the case, the optimal value for $\lambda$ must satisfy:

$$\lambda = \Big( \frac{\sum_{w \in OUT(i)} w g'_w(w/\lambda)}{\sum_{w \in IN(i)} w g'_w(\lambda w)} \Big)^{1/2} \tag{19}$$

If the regularizing function is the same for all the incoming and outgoing weights ($g_w = g$), then the optimal value $\lambda$ must satisfy:

$$\lambda = \Big( \frac{\sum_{w \in OUT(i)} w g'(w/\lambda)}{\sum_{w \in IN(i)} w g'(\lambda w)} \Big)^{1/2} \tag{20}$$

In particular, if $g(w) = |w|^p$ then $g(w)$ is differentiable except possibly at 0 and $g'(w) = s(w)p|w|^{p-1}$, where $s(w)$ denotes the sign of the weight $w$. Substituting in Equation 20, the optimal rescaling $\lambda$ must satisfy:

$$\lambda^* = \Big( \frac{\sum_{w \in OUT(i)} w s(w)|w|^{p-1}}{\sum_{w \in IN(i)} w|w s(w)|^{p-1}} \Big)^{1/2p} =$$
$$\Big( \frac{\sum_{w \in OUT(i)} |w|^p}{\sum_{w \in IN(i)} |w|^p} \Big)^{1/2p} = \Big( \frac{||OUT(i)||_p}{||IN(i)||_p} \Big)^{1/2} \tag{21}$$

At the optimum, no further balancing is possible, and thus $\lambda^* = 1$. Equation 18 yields immediately the generalized balance equation to be satisfied at the optimum:

$$\sum_{w \in IN(i)} w g'(w) = \sum_{w \in OUT(i)} w g'(w) \tag{22}$$

In the case of $L_P$ regularization, it is easy to check by applying Equation 22, or by direct calculation that:

$$\sum_{w \in IN(i)} |\lambda^* w|^p = \sum_{w \in OUT(i)} |w/\lambda^*|^p \tag{23}$$

which is the generalized balance equation. Thus after balancing neuron, the weights of neuron $i$ satisfy the $L_p$ balance (Equation 15). The change in the value of the regularizer is given by:

$$\Delta R = \sum_{w \in IN(i)} |w|^p + \sum_{w \in OUT(i)} |w|^p - \sum_{w \in IN(i)} |\lambda^* w|^p - \sum_{w \in OUT(i)} |w/\lambda^*|^p \tag{24}$$

By substituting $\lambda^*$ by its explicit value given by Equation 21 and collecting terms gives Equation 14. □

**Remark C.27.** *The monotonicity of the functions $g_w$ is not needed to prove the first part of Theorem C.26. It is only needed to prove uniqueness of $\lambda^*$ in the $L_p$ cases.*

**Remark C.28.** *Note that the same approach applies to the case where there are multiple additive regularizers. For instance with both $L^2$ and $L^1$ regularization, in this case the function $f$ has the form: $g_w(w) = \alpha w^2 + \beta|w|$. Generalized balance still applies. It also applies to the case where different regularizers are applied in different disconnected portions of the network.*

**Remark C.29.** *The balancing of a single BiLU neuron has little to do with the number of connections. It applies equally to fully connected neurons, or to sparsely connected neurons.*

## C.5 Scaling and Balancing Beyond BiLU Activation Functions

So far we have generalized ReLU activation functions to BiLU activation functions in the context of scaling and balancing operations with positive scaling factors. While in the following sections we will continue to work with BiLU activation functions, in this section we show that the scaling and balancing operations can be extended even further to other activation functions. The section can be skipped if one prefers to progress towards the main results on stochastic balancing.

Given a neuron with activation function $f(x)$, during scaling instead of multiplying and dividing by $\lambda > 0$, we could multiply the incoming weights by a function $g(\lambda)$ and divide the outgoing weights by a function $h(\lambda)$, as long as the activation function $f$ satisfies:

$$f(g(\lambda)x) = h(\lambda)f(x) \tag{25}$$

for every $x \in \mathbb{R}$ to ensure that the contribution of the neuron to the rest of the network remains unchanged. Note that if the activation function $f$ satisfies Equation 25, so does the activation function $-f$. In Equation 25, $\lambda$ does not have to be positive–we will simply assume that $\lambda$ belongs to some open (potentially infinite) interval $(a, b)$. Furthermore, the functions $g$ and $h$ cannot be zero for $\lambda \in (a, b)$ since they are used for scaling. It is reasonable to assume that the functions $g$ and $h$ are continuous, and thus they must have a constant sign as $\lambda$ varies over $(a, b)$.

Now, taking $x = 0$ gives $f(0) = h(\lambda)f(0)$ for every $\lambda \in (a, b)$, and thus either $f(0) = 0$ or $h(\lambda) = 1$ for every $\lambda \in (a, b)$. The latter is not interesting and thus we can assume that the activation function $f$ satisfies $f(0) = 0$. Taking $x = 1$ gives $f(g(\lambda)) = h(\lambda)f(1)$ for every $\lambda$ in $(a, b)$. For simplicity, let us assume that $f(x) = 1$. Then, we have: $f(g(\lambda)) = h(\lambda)$ for every $\lambda$. Substituting in Equation 25 yields:

$$f(g(\lambda)x) = f(g(\lambda))f(x) \tag{26}$$

for every $x \in \mathbb{R}$ and every $\lambda \in (a, b)$. This relation is essentially the same as the relation that defines multiplicative activation functions over the corresponding domain (see Proposition C.4), and thus we can identify a key family of solutions using power functions. Note that we can define a new parameter $\mu = g(\lambda)$, where $\mu$ ranges also over some positive or negative interval $I$ over which: $f(\mu x) = f(\mu)f(x)$.

### C.5.1 Bi-Power Units (BiPU)

Let us assume that $\lambda > 0$, $g(\lambda) = \lambda$ and $h(\lambda) = \lambda^c$ for some $c \in \mathbb{R}$. Then the activation function must satisfy the equation:

$$f(\lambda x) = \lambda^c f(x) \tag{27}$$

for any $x \in \mathbb{R}$ and any $\lambda > 0$. Note that if $f(x) = x^c$ we get a multiplicative activation function. More generally, these functions are characterized by the following proposition.

**Proposition C.30.** *The set of activation functions $f$ satisfying $f(\lambda x) = \lambda^c f(x)$ for any $x \in \mathbb{R}$ and any $\lambda > 0$ consist of the functions of the form:*

$$f(x) = \begin{cases} Cx^c & \text{if} \quad x \geq 0 \\ Dx^c & \text{if} \quad x < 0. \end{cases} \tag{28}$$

*where $c \in \mathbb{R}$, $C = f(1) \in R$, and $D = f(-1) \in \mathbb{R}$. We call these bi-power units (BiPU). If, in addition, we want $f$ to be continuous at 0, we must have either $c > 0$, or $c = 0$ with $C = D$.*

Given the general shape, these activations functions can be called BiPU (Bi-Power-Units). Note that in the general case where $c > 0$, $C$ and $D$ do not need to be equal. In particular, one of them can be equal to zero, and the other one can be different from zero giving rise to "rectified power units" (Figure 13).

*Proof.* By taking $x = 1$, we get $f(\lambda) = f(1)\lambda^c$ for any $\lambda > 0$. Let $f(1) = C$. Then we see that for any $x > 0$ we must have: $f(x) = Cx^c$. In addition, for every $\lambda > 0$ we must have: $f(\lambda 0) = f(0) = \lambda^c f(0)$. So if $c = 0$, then $f(x) = C = f(1)$ for $x \geq 0$. If $c \neq 0$, then $f(0) = 0$. In

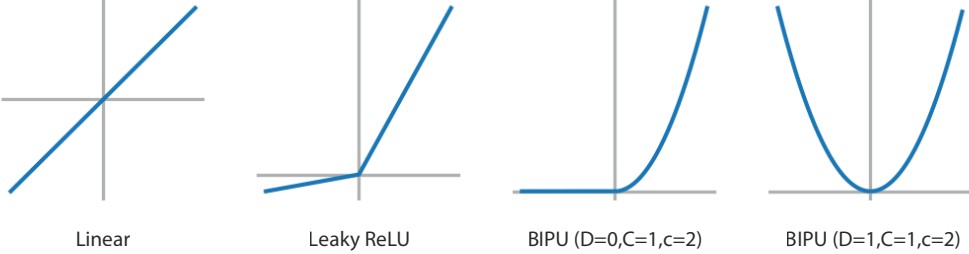



| | | | |
|---|---|---|---|
| Linear | Leaky ReLU | BIPU (D=0,C=1,c=2) | BIPU (D=1,C=1,c=2) |

Figure 13



this case, if we want the activation function to be continuous, then we see that we must have $c \geq 0$. So in summary for $x > 0$ we must have $f(x) = f(1)x^c = Cx^c$. For the function to be right continuous at 0, we must have either $f(0) = f(1) = C$ with $c = 0$ or $f(0) = 0$ with $c > 0$. We can now look at negative values of $x$. By the same reasoning, we have $f(\lambda(-1)) = f(-\lambda) = \lambda^c f(-1)$ for any $\lambda > 0$. Thus for any $x < 0$ we must have: $f(x) = f(-1)|x|^c = D|x|^c$ where $D = f(-1)$. Thus, if $f$ is continuous, there are two possibilities. If $c = 0$, then we must have $C = f(1) = D(f - 1)-$ and thus $f(x) = C$ everywhere. If $c \neq 0$, then continuity requires that $c > 0$. In this case $f(x) = Cx^c$ for $x \geq 0$ with $C = f(1)$, and $f(x) = Dx^c$ for $x < 0$ with $f(-1) = D$. In all cases, it is easy to check directly that the resulting functions satisfy the functional equation given by Equation 27. □

### C.5.2   Scaling BiPU Neurons

A BiPU neuron can be scaled by multiplying its incoming weight by $\lambda > 0$ and dividing its outgoing weights by $1/\lambda^c$. This will not change the role of the corresponding unit in the network, and thus it will not change the input-output function of the network.

### C.5.3   Balancing BiPU Neurons

As in the case of BiLU neurons, we balance a multiplicative neuron by asking what is the optimal scaling factor $\lambda$ that optimizes a particular regularizer. For simplicity, here we assume that the regularizer is in the $L_p$ class. Then we are interested in the value of $\lambda > 0$ that minimizes the function:

$$\lambda^p \sum_{w \in IN} |w|^p + \frac{1}{\lambda^{pc}} \sum_{w \in OUT} |w|^p \tag{29}$$

A simple calculation shows that the optimal value of $\lambda$ is given by:

$$\lambda^* = \left( \frac{c \sum_{OUT} |w|^p}{\sum_{IN} |w|^p} \right)^{1/p(c+1)} \tag{30}$$

Thus after balancing the weights, the neuron must satisfy the balance equation:

$$c \sum_{OUT} |w|^p = \sum_{IN} |w|^p \tag{31}$$

in the new weights $w$.

So far, we have focused on balancing individual neurons. In the next two sections, we look at balancing across all the units of a network. We first look at what happens to network balance when a network is trained by gradient descent and then at what happens to network balance when individual neurons are balanced iteratively in a regular or stochastic manner.

### C.6   Network Balance: Gradient Descent

A natural question is whether gradient descent (or stochastic gradient descent) applied to a network of BiLU neurons, with or without a regularizer, converges to a balanced state of the network, where all

the BiLU neurons are balanced. So we first consider the case where there is no regularizer ($\mathcal{E} = E$). The results in Du et al. [2018] may suggest that gradient descent may converge to a balanced state. In particular, they write that for any neuron $i$:

$$\frac{d}{dt}\Big( \sum_{w \in IN(i)} w^2 - \sum_{w \in OUT(i)} w^2 \Big) = 0 \tag{32}$$

Thus the gradient flow exactly preserves the difference between the $L_2$ cost of the incoming and outgoing weights or, in other words, the derivative of the $L_2$ balance *deficit* is zero. Thus if one were to start from a balanced state and use an infinitesimally small learning rate one ought to stay in a balanced state at all times.

However, it must be noted that this result was derived for the $L_2$ metric only, and thus would not cover other $L_p$ forms of balance. Furthermore, it requires an infinitesimally small learning rate. In practice, when any standard learning rate is applied, we find that gradient descent does *not* converge to a balanced state (Figure 1). However, things are different when a regularizer term is included in the error functions as described in the following theorem.

**Theorem C.31.** *Gradient descent in a network of BiLU units with error function $\mathcal{E} = E + R$ where $R$ has the properties described in Theorem C.26 (including all $L_p$) must converge to a balanced state, where every BiLU neuron is balanced.*

*Proof.* By contradiction, suppose that gradient descent converges to a state that is unbalanced and where the gradient with respect to all the weights is zero. Then there is at least one unbalanced neuron in the network. We can then multiply the incoming weights of such a neuron by $\lambda$ and the outgoing weights by $1/\lambda$ as in the previous section without changing the value of $E$. Since the neuron is not in balance, we can move $\lambda$ infinitesimally so as to reduce $R$, and hence $\mathcal{E}$. But this contradicts the fact that the gradient is zero. $\qquad\square$

**Remark C.32.** *In practice, in the case of stochastic gradient descent applied to $E + R$, at the end of learning the algorithm may hover around a balanced state. If the state reached by the stochastic gradient descent procedure is not approximately balanced, then learning ought to continue. In other words, the degree of balance could be used to monitor whether learning has converged or not. Balance is a necessary, but not sufficient, condition for being at the optimum.*

**Remark C.33.** *If early stopping is being used to control overfitting, there is no reason for the stopping state to be balanced. However, the balancing algorithms described in the next section could be used to balance this state.*

### C.7 Network Balance: Stochastic or Deterministic Balancing Algorithms

In this section, we look at balancing algorithms where, starting from an initial weight configuration $W$, the BiLU neurons of a network are balanced iteratively according to some deterministic or stochastic schedule that periodically visits all the neurons. We can also include algorithms where neurons are partitioned into groups (e.g. neuronal layers) and neurons in each group are balanced together.

#### C.7.1 Basic Stochastic Balancing

The most interesting algorithm is when the BiLU neurons of a network are iteratively balanced in a purely stochastic manner. This algorithm is particularly attractive from the standpoint of physically implemented neural networks because the balancing algorithm is local and the updates occur randomly without the need for any kind of central coordination. As we shall see in the following section, the random local operations remarkably lead to a unique form of global order. The proof for the stochastic case extends immediately to the deterministic case, where the BiLU neurons are updated in a deterministic fashion, for instance by repeatedly cycling through them according to some fixed order.

#### C.7.2 Subset Balancing (Independent or Tied)

It is also possible to partition the BiLU neurons into non-overlapping subsets of neurons, and then balance each subset, especially when the neurons in each subset are disjoint of each other. In this

case, one can balance all the neurons in a given subset, and repeat this subset-balancing operation subset-by-subset, again in a deterministic or stochastic manner. Because the BiLU neurons in each subset are disjoint, it does not matter whether the neurons in a given subset are updated synchronously or sequentially (and in which order). Since the neurons are balanced independently of each other, this can be called independent subset balancing. For example, in a layered feedforward network with no lateral connections, each layer corresponds to a subset of disjoint neurons. The incoming and outgoing connections of each neuron are distinct from the incoming and outgoing connections of any other neuron in the layer, and thus the balancing operation of any neuron in the layer does not interfere with the balancing operation of any other neuron in the same layer. So this corresponds to independent layer balancing,

As a side note, balancing a layer $h$, may disrupt the balance of layer $h + 1$. However, balancing layer $h$ and $h + 2$ (or any other layer further apart) can be done without interference of the balancing processes. This suggests also an alternating balancing scheme, where one alternatively balances all the odd-numbered layers, and all the evenly-numbered layers.

Yet another variation is when the neurons in a disjoint subset are tied to each other in the sense that they must all share the same scaling factor $\lambda$. In this case, balancing the subset requires finding the optimal $\lambda$ for the entire subset, as opposed to finding the optimal $\lambda$ for each neuron in the subset. Since the neurons are balanced in a coordinated or tied fashion, this can be called coordinated or tied subset balancing. For example, tied layer balancing must use the same $\lambda$ for all the neurons in a given layer. It is easy to see that this approach leads to layer synaptic balance which has the form (for an $L_p$ regularizer):

$$\sum_i \sum_{w \in IN(i)} |w|^p = \sum_i \sum_{w \in OUT(i)} |w|^p \tag{33}$$

where $i$ runs over all the neurons in the layer. This does *not* necessarily imply that each neuron in the layer is individually balanced. Thus neuronal balance for every neuron in a layer implies layer balance, but the converse is not true. Independent layer balancing will lead to layer balance. Coordinated layer balancing will lead to layer balance, but not necessarily to neuronal balance of each neuron in the layer. Layer-wise balancing, independent or tied, can be applied to all the layers and in deterministic (e.g. sequential) or stochastic manner. Again the proof given in the next section for the basic stochastic algorithm can easily be applied to these cases (see also Appendix B).

### C.7.3   Remarks about Weight Sharing and Convolutional Neural Networks

Suppose that two connections share the same weight so that we must have: $w_{ij} = w_{kl}$ at all times. In general, when the balancing algorithm is applied to neuron $i$ or $j$, the weight $w_{ij}$ will change and the same change must be applied to $w_{kl}$. The latter may disrupt the balance of neuron $k$ or $l$. Furthermore, this may not lead to a decrease in the overall value of the regularizer $R$.

The case of convolutional networks is somewhat special, since *all* the incoming weights of the neurons sharing the same convolutional kernel are shared. However, in general, the outgoing weights are not shared. Furthermore, certain operations like max-pooling are not homogeneous. So if one trains a CNN with $E$ alone, or even with $E + R$, one should not expect any kind of balance to emerge in the convolution units. However, all the other BiLU units in the network should become balanced by the same argument used for gradient descent above. The balancing algorithm applied to individual neurons, or the independent layer balancing algorithm, will not balance individual neurons sharing the same convolution kernel. The only balancing algorithm that could lead to some convolution layer balance, but not to individual neuronal balance, is the coordinated layer balancing, where the same $\lambda$ is used for all the neurons in the same convolution layer, provided that their activation functions are BiLU functions.

We can now study the convergence properties of balancing algorithms.

### C.8   Convergence of Balancing Algorithms

We now consider the basic stochastic balancing algorithm, where BiLU neurons are iteratively and stochastically balanced. It is essential to note that balancing a neuron $j$ may break the balance of another neuron $i$ to which $j$ is connected. Thus convergence of iterated balancing is not obvious.

There are three key questions to be addressed for the basic stochastic algorithm, as well as all the other balancing variations. First, does the value of the regularizer converges to a finite value? Second, do the weights themselves converge to fixed finite values representing a balanced state for the entire network? And third, if the weights converge, do they always converge to the same values, irrespective of the order in which the units are being balanced? In other words, given an initial state $W$ for the network, is there a unique corresponding balanced state, with the same input-output functionalities?

### C.8.1 Notation and Key Questions

For simplicity, we use a continuous time notation. After a certain time $t$ each neuron has been balanced a certain number of times. While the balancing operations are not commutative as balancing operations, they are commutative as scaling operations. Thus we can reorder the scaling operations and group them neuron by neuron so that, for instance, neuron $i$ has been scaled by the sequence of scaling operations:

$$S_{\lambda_1^*}(i) S_{\lambda_2^*}(i) \ldots S_{\lambda_{n_{it}}^*}(i) = S_{\Lambda_i(t)}(i) \tag{34}$$

where $n_{it}$ corresponds to the count of the last update of neuron $i$ prior to time $t$, and:

$$\Lambda_i(t) = \prod_{1 \leq n \leq n_{it}} \lambda_n^*(i) \tag{35}$$

For the input and output units, we can consider that their balancing coefficients $\lambda^*$ are always equal to 1 (at all times) and therefore $\Lambda_i(t) = 1$ for any visible unit $i$.

Thus, we first want to know if $R$ converges. Second, we want to know if the weights converge. This question can be split into two sub-questions: (1) Do the balancing factors $\lambda_n^*(i)$ converge to a limit as time goes to infinity. Even if the $\lambda_n^*(i)$'s converge to a limit, this does not imply that the weights of the network converge to a limit. After a time $t$, the weight $w_{ij}(t)$ between neuron $j$ and neuron $i$ has the value $w_{ij}\Lambda_i(t)/\Lambda_j(t)$, where $w_{ij} = w_{ij}(0)$ is the value of the weight at the start of the stochastic balancing algorithm. Thus: (2) Do the quantities $\Lambda_i(t)$ converge to finite values, different from 0? And third, if the weights converge to finite values different from 0, are these values unique or not, i.e. do they depend on the details of the stochastic updates or not? These questions are answered by the following main theorem..

### C.8.2 Convergence of the Basic Stochastic Balancing Algorithm to a Unique Optimum

**Theorem C.34.** *(Convergence of Stochastic Balancing) Consider a network of BiLU neurons with an error function $\mathcal{E}(W) = E(W) + R(W)$ where $R$ satisfies the conditions of Theorem C.22 including all $L_p$ ($p > 0$). Let $W$ denote the initial weights. When the neuronal stochastic balancing algorithm is applied throughout the network so that every neuron is visited from time to time, then $E(W)$ remains unchanged but $R(W)$ must converge to some finite value that is less or equal to the initial value, strictly less if the initial weights are not balanced. In addition, for every neuron $i$, $\lambda_i^*(t) \to 1$ and $\Lambda_i(t) \to \Lambda_i$ as $t \to \infty$, where $\Lambda_i$ is finite and $\Lambda_i > 0$ for every $i$. As a result, the weights themselves must converge to a limit $W'$ which is globally balanced, with $E(W) = E(W')$ and $R(W) \geq R(W')$, and with equality if only if $W$ is already balanced. Finally, $W'$ is unique as it corresponds to the solution of a strictly convex optimization problem in the variables $L_{ij} = \log(\Lambda_i/\Lambda_j)$ with linear constraints of the form $\sum_\pi L_{ij} = 0$ along any path $\pi$ joining an input unit to an output unit and along any directed cycle (for recurrent networks). Stochastic balancing projects to stochastic trajectories in the linear manifold that run from the origin to the unique optimal configuration.*

*Proof.* Each individual balancing operation leaves $E(W)$ unchanged because the BiLU neurons are homogeneous. Furthermore, each balancing operation reduces the regularization error $R(W)$, or leaves it unchanged. Since the regularizer is lower-bounded by zero, the value of the regularizer must approach a limit as the stochastic updates are being applied.

For the second question, when neuron $i$ is balanced at some step, we know that the regularizer $R$ decreases by:

$$\Delta R = \left( \left( \sum_{w \in IN(i)} |w|^p \right)^{1/2} - \left( \sum_{w \in OUT(i)} |w|^p \right)^{1/2} \right)^2 \tag{36}$$

If the convergence were to occur in a finite number of steps, then the coefficients $\lambda_i^*(t)$ must become equal and constant to 1 and the result is obvious. So we can focus on the case where the convergence does not occur in a finite number of steps (indeed this is the main scenario, as we shall see at the end of the proof). Since $\Delta R \to 0$, we must have:

$$\sum_{w \in IN(i)} |w|^p \to \sum_{w \in OUT(i)} |w|^p \tag{37}$$

But from the expression for $\lambda^*$ (Equation 21), this implies that for every $i$, $\lambda_n^*(i) \to 1$ as time increases ($n \to \infty$). This alone is not sufficient to prove that $\Lambda_i(t)$ converges for every $i$ as $t \to \infty$. However, it is easy to see that $\Lambda_i(t)$ cannot contain a sub-sequence that approaches 0 or $\infty$ (Figure 14). Furthermore, not only $\Delta R$ converges to 0, but the series $\sum \Delta R$ is convergent. This shows that, for every $i$, $\Delta_i(t)$ must converge to a finite, non-zero value $\Delta_i$. Therefore all the weights must converge to fixed values given by $w_{ij}(0)\Lambda_i/\Lambda_j$.

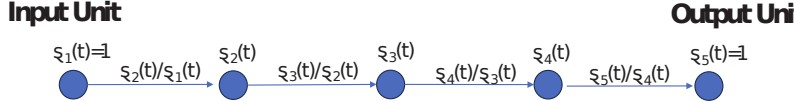

Figure 14: A path with three hidden BiLU units connecting one input unit to one output unit. During the application of the stochastic balancing algorithm, at time $t$ each unit $i$ has a cumulative scaling factor $\Lambda_i(t)$, and each directed edge from unit $j$ to unit $i$ has a scaling factor $M_{ij}(t) = \Lambda_i(t)/\Lambda_j(t)$. The $\lambda_i(t)$ must remain within a finite closed interval away from 0 and infinity. To see this, imagine for instance that there is a subsequence of $\Lambda_3(t)$ that approaches 0. Then there must be a corresponding subsequence of $\Lambda_4(t)$ that approaches 0, or else the contribution of the weight $w_{43}\Lambda_4(t)/\Lambda_3(t)$ to the regularizer would go to infinity. But then, as we reach the output layer, the contribution of the last weight $w_{54}\Lambda_5(t)/\Lambda_4(t)$ to the regularizer goes to infinity because $\Lambda_5(t)$ is fixed to 1 and cannot compensate for the small values of $\Lambda_4(t)$. And similarly, if there is a subsequence of $\Lambda_3(t)$ going to infinity, we obtain a contradiction by propagating its effect towards the input layer.

Finally, we prove that given an initial set of weights $W$, the final balanced state is unique and independent of the order of the balancing operations. The coefficients $\Lambda_i$ corresponding to a globally balanced state must be solutions of the following optimization problem:

$$\min_{\Lambda} R(\Lambda) = \sum_{ij} |\frac{\Lambda_i}{\Lambda_j} w_{ij}|^p \tag{38}$$

under the simple constraints: $\Lambda_i > 0$ for all the BiLU hidden units, and $\Lambda_i = 1$ for all the visible (input and output) units. In this form, the problem is not convex. Introducing new variables $M_j = 1/\Lambda_j$ is not sufficient to render the problem convex. Using variables $M_{ij} = \Lambda_i/\Lambda_j$ is better, but still problematic for $0 < p \le 1$. However, let us instead introduce the new variables $L_{ij} = \log(\Lambda_i/\Lambda_j)$. These are well defined since we know that $\Lambda_i/\Lambda_j > 0$. The objective now becomes:

$$\min R(L) = \sum_{ij} |e^{L_{ij}} w_{ij}|^p = \sum_{ij} e^{pL_{ij}} |w_{ij}|^p \tag{39}$$

This objective is strictly convex in the variables $L_{ij}$, as a sum of strictly convex functions (exponentials). However, to show that it is a convex optimization problem we need to study the constraints

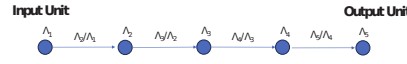

Figure 15: A path with five units. After the stochastic balancing algorithm has converged, each unit $i$ has a scaling factor $\Lambda_i$, and each directed edge from unit $j$ to unit $i$ has a scaling factor $M_{ij} = \Lambda_i/\Lambda_j$. The products of the $M_{ij}$'s along the path is given by: $\frac{\Lambda_2}{\Lambda_1} \frac{\Lambda_3}{\Lambda_2} \frac{\Lambda_4}{\Lambda_3} \frac{\Lambda_5}{\Lambda_4} = \frac{\Lambda_5}{\Lambda_1}$. Accordingly, if we sum the variables $L_{ij} = \log M_{ij}$ along the directed path, we get $L_{21} + L_{32} + L_{43} + L_{54} = \log \Lambda_5 - \log \Lambda_1$. In particular, if unit 1 is an input unit and unit 5 is an output unit, we must have $\Lambda_1 = \Lambda_5 = 1$ and thus: $L_{21} + L_{32} + L_{43} + L_{54} = 0$. Likewise, in the case of a directed cycle where unit 1 and unit 5 are the same, we must have: $L_{21} + L_{32} + L_{43} + L_{54} + L_{15} = 0$.

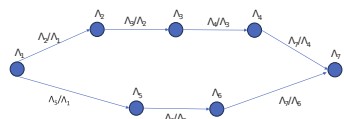

Figure 16: Two hidden units (1 and 7) connected by two different directed paths 1-2-3-4-7 and 1-5-6-7 in a BiLU network. Each unit $i$ has a scaling factor $\Lambda_i$, and each directed edge from unit $j$ to unit $i$ has a scaling factor $M_{ij} = \Lambda_i/\Lambda_j$. The products of the $M_{ij}$'s along each path is equal to: $\frac{\Lambda_2}{\Lambda_1} \frac{\Lambda_3}{\Lambda_2} \frac{\Lambda_4}{\Lambda_3} \frac{\Lambda_7}{\Lambda_4} = \frac{\Lambda_5}{\Lambda_1} \frac{\Lambda_6}{\Lambda_5} \frac{\Lambda_7}{\Lambda_6} = \frac{\Lambda_7}{\Lambda_1}$. Therefore the variables $L_{ij} = \log M_{ij}$ must satisfy the linear equation: $L_{21} + L_{32} + L_{43} + L_{74} = L_{51} + L_{65} + L_{76} = \log \Lambda_7 - \log \Lambda_1$.

on the variables $L_{ij}$. In particular, from the set of $\Lambda_i$'s it is easy to construct a unique set of $L_{ij}$. However what about the converse?

**Definition C.35.** *A set of real numbers $L_{ij}$, one per connection of a given neural architecture, is self-consistent if and only if there is a unique corresponding set of numbers $\Lambda_i > 0$ (one per unit) such that: $\Lambda_i = 1$ for all visible units and $L_{ij} = \log \Lambda_i/\Lambda_j$ for every directed connection from a unit $j$ to a unit $i$.*

**Remark C.36.** *This definition depends on the graph of connections, but not on the original values of the synaptic weights. Every balanced state is associated with a self-consistent set of $L_{ij}$, but not every self-consistent set of $L_{ij}$ is associated with a balanced state.*

**Proposition C.37.** *A set $L_{ij}$ associated with a neural architecture is self-consistent if and only if $\sum_\pi L_{ij} = 0$ where $\pi$ is any directed path connecting an input unit to an output unit or any directed cycle (for recurrent networks).*

**Remark C.38.** *Thus the constraints associated with being a self-consistent configuration of $L_{ij}$'s are all linear. This resulting linear manifold $\mathcal{L}$ depends only on the architecture, i.e., the graph of connections, but not on the actual weight values. The strictly convex function $R(L_{ij})$ depends on the actual weights $W$. Different sets of weights $W$ produce different convex functions over the same linear manifold. If $E$ denotes the total number of connections, then obviously $\dim \mathcal{L} \le E$. In order to infer all the $\Lambda_i$, there must exist at least one constrained path going through each node $i$. Thus, in a layered feedforward network, the dimension of $\mathcal{L}$ is given by: $\dim \mathcal{L} = E - M$, where here $M$ denotes the size of the largest layer.*

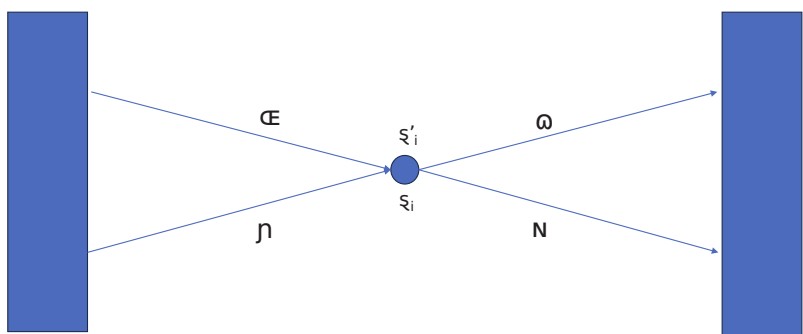

Figure 17: Consider two paths $\alpha + \beta$ and $\gamma + \delta$ from the input layer to the output layer going through the same unit $i$. Let us assume that the first path assigns a multiplier $\Lambda_i$ to unit $i$ and the second path assigns a multiplier $\Lambda'_i$ to the same unit. By assumption we must have: $\sum_\alpha L_{ij} + \sum_\beta L_{ij} = 0$ for the first path, and $\sum_\gamma L_{ij} + \sum_\delta L_{ij} = 0$. But $\alpha + \delta$ and $\gamma + \beta$ are also paths from the input layer to the output layer and therefore: $\sum_\alpha L_{ij} + \sum_\delta L_{ij} = 0$ and $\sum_\gamma L_{ij} + \sum_\beta L_{ij} = 0$. As a result, $\sum_\alpha L_{ij} = \log \Lambda_i = \sum_\gamma L_{ij} = \Lambda'_i$. Therefore the assignment of the multiplier $\Lambda_i$ must be consistent across different paths going through unit $i$.

**Remark C.39.** *One could coalesce all the input units and all output units into a single unit, in which case a path from an input unit to and output unit becomes also a directed cycle. In this representation, the constraints are that the sum of the $L_{ij}$ must be zero along any directed cycle. In general, it is not necessary to write a constraint for every path from input units to output units. It is sufficient to select a representative set of paths such that every unit appears in at least one path.*

*Proof.* If we look at any directed path $\pi$ from unit $i$ to unit $j$, it is easy to see that we must have:

$$\sum_\pi L_{kl} = \log \Lambda_i - \log \Lambda_j \tag{40}$$

This is illustrated in Figures 15 and 16. Thus along any directed path that connects any input unit to any output unit, we must have $\sum_\pi L_{ij} = 0$. In addition, for recurrent neural networks, if $\pi$ is a directed cycle we must also have: $\sum_\pi L_{ij} = 0$. Thus in short we only need to add linear constraints of the form: $\sum_\pi L_{ij} = 0$. Any unit is situated on a path from an input unit to an output unit. Along that path, it is easy to assign a value $\Lambda_i$ to each unit by simple propagation starting from the input unit which has a multiplier equal to 1. When the propagation terminates in the output unit, it terminates consistently because the output unit has a multiplier equal to 1 and, by assumption, the sum of the multipliers along the path must be zero. So we can derive scaling values $\Lambda_i$ from the variables $L_{ij}$. Finally, we need to show that there are no clashes, i.e. that it is not possible for two different propagation paths to assign different multiplier values to the same unit $i$. The reason for this is illustrated in Figure 17. $\square$

We can now complete the proof Theorem C.34. Given a neural network of BiLUs with a set of weights $W$, we can consider the problem of minimizing the regularizer $R(L_{ij})$ over the self-admissible configuration $L_{ij}$. For any $p > 0$, the $L_p$ regularizer is strictly convex and the space of self-admissible configurations is linear and hence convex. Thus this is a strictly convex optimization problem that has a unique solution (Figure 18). Note that the minimization is carried over self-consistent configurations, which in general are not associated with balanced states. However, the configuration of the weights associated with the optimum set of $L_{ij}$ (point $A$ in Figure 18) must be balanced. To see this, imagine

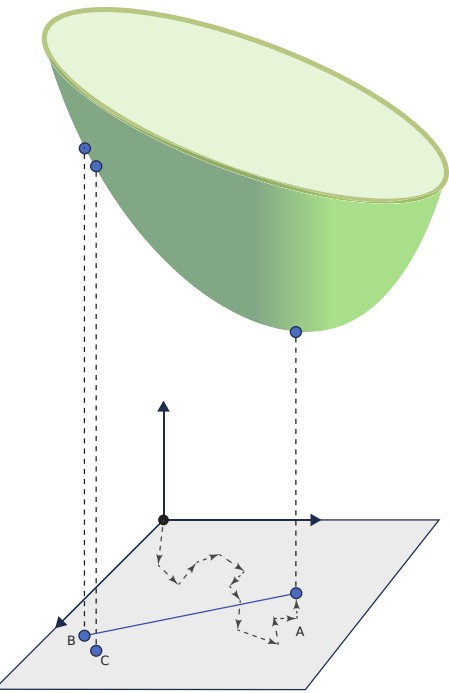

Figure 18: The problem of minimizing the strictly convex regularizer $R(L_{ij}) = \sum_{ij} e^{pL_{ij}} |w_{ij}|^p$ ($p > 0$), over the linear (hence convex) manifold of self-consistent configurations defined by the linear constraints of the form $\sum_{\pi} L_{ij} = 0$, where $\pi$ runs over input-output paths. The regularizer function depends on the weights. The linear manifold depends only on the architecture, i.e., the graph of connections. This is a strictly convex optimization problem with a unique solution associated with the point $A$. At $A$ the corresponding weights must be balanced, or else a self-consistent configuration of lower cost could be found by balancing any non-balanced neuron. Finally, any other self-consistent configuration $B$ cannot correspond to a balanced state of the network, since there must exist balancing moves that further reduce the regularizer cost (see main text). Stochastic balancing produces random paths from the origin, where $L_{ij=} \log M_{ij} = 0$, to the unique optimum point $A$.

that one of the BiLU units–unit $i$ in the network is not balanced. Then we can balance it using a multiplier $\lambda_i^*$ and replace $\Lambda_i$ by $\Lambda_i' = \Lambda_i \lambda^*$. It is easy to check that the new configuration including $\Lambda_i'$ is self-consistent. Thus, by balancing unit $i$, we are able to reach a new self-consistent configuration with a lower value of $R$ which contradicts the fact that we are at the global minimum of the strictly convex optimization problem.

We know that the stochastic balancing algorithm always converges to a balanced state. We need to show that it cannot converge to any other balanced state, and in fact that the global optimum is the only balanced state. By contradiction, suppose it converges to a different balanced state associated with the coordinates $(L_{ij}^B)$ (point $B$ in Figure 18). Because of the self-consistency, this point is also associated with a unique set of $(\Lambda_i^B)$ coordinates. The cost function is continuous and differentiable in both the $L_{ij}$'s and the $\Lambda_i$'s coordinates. If we look at the negative gradient of the regularizer, it is non-zero and therefore it must have at least one non-zero component $\partial R / \partial \Lambda_i$ along one of the $\Lambda_i$ coordinates. This implies that by scaling the corresponding unit $i$ in the network, the regularizer can be further reduced, and by balancing unit $i$ the balancing algorithm will reach a new point ($C$ in Figure 18) with lower regularizer cost. This contradicts the assumption that $B$ was associated with a balanced stated. Thus, given an initial set of weights $W$, the stochastic balancing algorithm must always converge to the same and unique optimal balanced state $W^*$ associated with the self-consistent point $A$. A particular stochastic schedule corresponds to a random path within the linear manifold from the origin (at time zero all the multipliers are equal to 1, and therefore for any $i$ and any $j$: $M_{ij} = 1$ and $L_{ij} = 0$) to the unique optimum point $A$. $\qquad\square$

**Remark C.40.** *From the proof, it is clear that the same result holds also for any deterministic balancing schedule, as well as for tied and non-tied subset balancing, e.g., for layer-wise balancing*

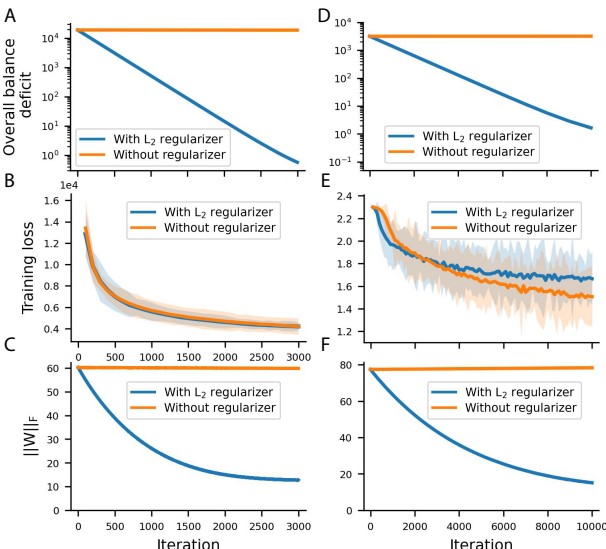

Figure 19: **SGD applied to $E$ alone, in general, does not converge to a balanced state, but sGD applied to $E + R$ converges to a balanced state. (A-C)** Simulations use a deep fully connected autoencoder trained on the MNIST dataset. **(D-F)** Simulations use a deep locally connected network trained on the CIFAR10 dataset. **(A,D)** Regularization leads to neural balance. **(B,E)** The training loss decreases and converges during training (these panels are not meant for assessing the quality of learning when using a regularizer). **(C,F)** Using weight regularization decreases the norm of weights. **(A-F)** Shaded areas correspond to one s.t.d around the mean (in some cases the s.t.d. is small and the shaded area is not visible).

*and tied layer-wise balancing. In the Appendix, we provide an analytical solution for the case of tied layer-wise balancing in a layered feed-forward network.*

**Remark C.41.** *The same convergence to the unique global optimum is observed if each neuron, when stochastically visited, is partially balanced (or favorably scaled) rather than fully balanced, i.e., it is scaled with a factor that reduces $R$ but not necessarily minimizes $R$. Stochastic balancing can also be viewed as a form of EM algorithm where the E and M steps can be taken fully or partially.*

### C.8.3 Convergence to a Unique Optimum for BiPU Stochastic Balancing

We have seen that a generalized form of scaling and balancing can be defined for more general units than BiLUs, in particular for BiPUs. Thus now we consider a network of units with activations functions $f$ satisfying the relationship: $f(\lambda x) = \lambda^c f(x)$ (note that this includes BiLU units for $c = 1$). We even allow $c$ to vary from unit to unit.

It is easy to see that most of the analyses above done for BiLU units apply to this generalization. In particular, if we apply stocahstic generalized balancing, in the limit the positive multipliers of each connection $w_{ij}$ must satisfy:

$$M_{ij} = \Lambda_i / \Lambda_j^{c_j} \tag{41}$$

As above, we can define a new set of variables $L_{ij} = \log M_{ij}$ and, for any $p > 0$, the regularizer $R(L) = \sum_{ij} e^{pL_{ij}} |w_{ij}|^p$ is strictly convex. What is different, however, is the set of constraints on the variables $L_{ij}$. These are the constraints that allow one to compute the variables $\Lambda_i$ uniquely from the variables $L_{ij}$ (or, equivalently, the variables $M_{ij}$). This is addressed by the following theorem.

**Theorem C.42.** *Under the same conditions of Theorem C.34, but using activation functions that satisfy for each unit $i$ the relationship $f(\lambda x) = \lambda^{c_i} f(x)$, the corresponding stochastic generalized balancing algorithm converges to the unique minimum of a strictly convex optimization problem in the variables $L_{ij}$. The strictly convex objective function is given by $R(L) = \sum_{ij} e^{pL_{ij}} |w_{ij}|^p$. The constraints are linear and of the form:*

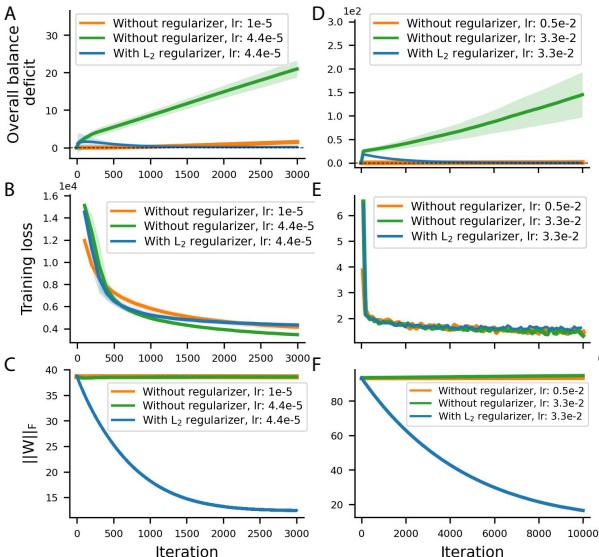

Figure 20: **Even if the starting state is balanced, SGD does not preserve the balance unless the learning rate is infinitely small.** **(A-C)** Simulations use a deep fully connected autoencoder trained on the MNIST dataset. **(D-F)** Simulations use deep locally connected network trained on the CIFAR10 dataset. **(A-F)** The initial weights are balanced using the stochastic balancing algorithm. Then the network is trained by SGD. **(A,D)** When the learning rate (lr) is relatively large, without regularization, the initial balance of the network is rapidly disrupted. **(B,E)** The training loss decreases and converges during training (these panels are not meant for assessing the quality of learning when using a regularizer). **(C,F)** Using weight regularization decreases the norm of the weights. **(A-F)** Shaded areas correspond to one s.t.d around the mean (in some cases the s.t.d. is small and the shaded area is not visible).

$$\sum_{i \in \pi} \left( \prod_{k=i}^{n} c_k \right) L_{ii-1} = 0 \tag{42}$$

*for each path $\pi$ from an input unit to an output unit, going sequentially through the units $0, 1, \ldots, n$, where 0 corresponds to the input unit, and $n$ corresponds to the output unit of the path. The set of paths in the constraints must cover all the units in the network.*

*Proof.* Let us assume that there is a consistent set of multipliers $\Lambda_0, \ldots, \Lambda_n$ associated with the coefficients $L_{ii-1} = \log M_{ii-1}$ along the path $\pi$, with $\Lambda_0 = \Lambda_n = 1$. Since $M_{ii-1} = \Lambda_i / \Lambda_{i-1}^{c_{i-1}}$, we can derive the multipliers $\Lambda_i$ iteratively by propagating information from the input unit to the output unit, in the form:

$$\Lambda_i = M_{ii-1}\Lambda_{i-1}^{c_{i-1}} \quad \text{or} \quad \log \Lambda_i = L_{ii-1} + c_{i-1} \log \Lambda_{i-1} \tag{43}$$

Using the boundary conditions $\Lambda_0 = \Lambda_n = 1$ gives the formula in Theorem C.42. The same arguments given for BiLU units can be used to complete the proof. $\qquad\square$

**Remark C.43.** *Note that if all the units have the same exponent $c$ associated with the scaling of their activation functions, then the linear constraints have the simplified form:*

$$\sum_{i \in \pi} c^{n+1-i} L_{ii-1} = 0 \tag{44}$$

## Universal Approximation Properties of BiLU Neurons

Here we show that any continuous real-valued function defined over a compact set of the Euclidean space can be approximated to any degree of precision by a network of BiLU neurons with a single hidden layer. As in the case of the similar proof given in Baldi [2021] using linear threshold gates in the hidden layer, it is enough to prove the theorem for a continuous function $f \colon 0, 1 \to \mathbb{R}$.

**Theorem C.44.** *(Universal Approximation Properties of BiLU Neurons) Let $f$ be any continuous function from $[0, 1]$ to $\mathbb{R}$ and $\epsilon > 0$. Let $g_\lambda$ be the ReLU activation function with slope $\lambda \in \mathbb{R}s$. Then there exists a feedforward network with a single hidden layer of neurons with ReLU activations of the form $g_\lambda$ and a single output linear neuron, i.e., with BiLU activation equal to the identity function, capable of approximating $f$ everywhere within $\epsilon$ (sup norm).*

*Proof.* To be clear, $g_\lambda(x) = 0$ for $x < 0$ and $g_\lambda(x) = \lambda x$ for $0 \leq x$. Since $f$ is continuous over a compact set, it is uniformly continuous. Thus there exists $\alpha > 0$ such that for any $x_1$ and $x_2$ in the $[0, 1]$ interval:

$$|x_2 - x_1| < \alpha \implies |f(x_2) - f(x_1)| < \epsilon \tag{45}$$

Let $N$ be an integer such that $1 < N\alpha$, and let us slice the interval $[0, 1]$ into $N$ consecutive slices of width $h = 1/N$, so that within each slice the function $f$ cannot jump by more than $\epsilon$. Let us connect the input unit to all the hidden units with a weight equal to 1. Let us have $N$ hidden units numbered $1, \ldots, N$ with biases equal to $0, 1/N, 2/N, \ldots, N_1/N$ respectively and activation function of the form $g_{\lambda_k}$. It is essential that different units be allowed to have different slopes $\lambda_k$. The input unit is connected to all the hidden units and all the weights on these connections are equal to 1. Thus when $x$ is in the $k$-th slice, $(k-1)/N \leq x < k/N$, all the units from $k+1$ to $N$ have an output equal to 0, and all the units from 1 to $k$ have an output determined by the corresponding slopes. All the hidden units are connected to the output unit with weights $\beta_1, \ldots, \beta_N$, and $\beta_0$ is the bias of the output unit. We want the output unit to be linear. In order for the $\epsilon$ approximation to be satisfied, it is sufficient if in the $(k-1)/N \leq x < k/N$ interval, the output is equal to the line joining the point $f((k-1)/N)$ to the point $f(k/N)$. In other words, if $x \in [(k-1)/N, k/N)$, then we want the output of the network to be:

$$\beta_0 + \sum_{i=1}^{k} \beta_i \lambda_i (x - (i-1)h) =$$
$$f(\frac{k-1}{N}) + \frac{f(\frac{k}{N}) - f(\frac{k-1}{N})}{h}(x - (k-1)h) \tag{46}$$

By equating the y-intercept and slope of the lines on the left-hand side and the righ- hand side of Equation 46, we can solve for the weights $\beta$'s and the slopes $\lambda$'s. $\qquad \square$

As in the case of the similar proof using linear threshold functions in the hidden layer (see Baldi [2021],) this proof can easily be adapted to continuous functions defined over a compact set of $\mathbb{R}^n$, even with a finite number of finite discontinuities, and into $\mathbb{R}^m$.

## Analytical Solution for the Unique Global Balanced State

Here we directly prove the convergence of stochastic balancing to a unique final balanced state, and derive the equations for the balanced state, in the special case of tied layer balancing (as opposed to single neuron balancing). The Proof and the resulting equations are also valid for stochastic balancing (one neuron at a time) in a layered architecture comprising a single neuron per layer. Let us call tied layer scaling the operation by which all the incoming weights to a given layer of BiLU neurons are multiplied by $\lambda > 0$ and all the outgoing weights of the layer are multiplied by $1/\lambda$, again leaving the training error unchanged. Let us call layer balancing the particular scaling operation corresponding to the value of $\lambda$ that minimizes the contribution of the layer to the $L_2$ (or any other $L_p$) regularizer vaue. This optimal value of $\lambda^*$ results in layer-wise balance equations: the sum of the squares of all the incoming weights of the layer must be equal to the sum of the squares of all the outgoing weights of the layer in the $L_2$ case, and similarly in all $L^P$ cases.

**Theorem C.45.** *Assume that tied layer balancing is applied iteratively and stochastically to the layers of a layered feedforward network of BiLU neurons. As long as all the layers are visited periodically, this procedure will always converge to the same unique set of weights, which will satisfy the layer-balance equations at all layers, irrespective of the details of the schedule. Furthermore, the balance state can be solved analytically.*

*Proof.* Every time a layer balancing operation is applied, the training error remains the same, and the $L_2$ (or any other $L_p$) regularization error decreases or stays the same. Since the regularization error is always positive, it must converge to a certain value. Using the same arguments as in the proof of Theorem C.34, the weights must also converge to a stable configuration, and since the configuration is stable all its layers must satisfy the layer-wise balance equation. The key remaining question is why is this configuration unique and can we solve it analytically? Let $A_1, A_2, \ldots A_N$ denote the matrices of connections between the layers of the network. Let $\Lambda_1, \Lambda_2, \ldots, \Lambda_{N-1}$ be $N-1$ strictly positive multipliers, representing the limits of the products of the corresponding $\lambda_i^*$ associated with each balancing step at layer $i$, as in the proof of Theorem C.34. In this notation, layer 0 is the input layer and layer $N$ is the output layer (with $\Lambda_0 = 1$ and $\Lambda_N = 1$).

After converging, each matrix $A_i$ becomes the matrix $\Lambda_i/\Lambda_{i-1}A_i = M_i A_i$ for $i = 1 \ldots N$, with $M_i = \lambda_i/\Lambda_{i-1}$. The multipliers $M_i$ must minimize the regularizer while satisfying $M_1 \ldots M_N = 1$ to ensure that the training error remains unchanged. In other words, to find the values of the $M_i$'s we must minimize the Lagrangian:

$$\mathcal{L}(M_1, \ldots, M_N) = \sum_{i=1}^{N} ||M_i A_i||^2 + \mu(1 - \prod_{i=1}^{N} M_i) \tag{47}$$

written for the $L^2$ case in terms of the Frobenius norm, but the analysis is similar in the general $L_p$ case. From this, we get the critical equations:

$$\frac{\partial \mathcal{L}}{\partial M_i} = 2M_i ||A_i||^2 - \mu M_1 \ldots M_{i-1} M_{i+1} \ldots M_N = 0$$

$$\text{for } i = 1, \ldots, N \quad \text{and} \quad \prod_{i=1}^{N} M_i = 1 \tag{48}$$

As a resut, for every $i$:

$$2M_i ||A_i||^2 - \frac{\mu}{M_i} = 0 \quad \text{or} \quad \mu = 2M_i^2 ||A_i||^2 \tag{49}$$

Thus each $M_i > 0$ can be expressed in a unique way as a function of the Lagrangian multiplier $\mu$ as: $M_i = (\mu/2||A_i||^2)^{1/2}$. By writing again that the product of the $M_i$ is equal to 1, we finally get:

$$\mu^N = 2^N \prod_{i=1}^{N} ||A_i||^2 \quad \text{or} \quad \mu = 2 \prod_{i=1}^{N} ||A_i||^{2/N} \tag{50}$$

Thus we can solve for $M_i$:

$$M_i = \frac{\mu}{2||A_i||^2} = \frac{\prod_{i=1}^{N} ||A_i||^{2/N}}{||A_i||^2} \quad \text{for } i = 1, \ldots, N \tag{51}$$

Thus, in short, we obtain a unique closed-form expression for each $M_i$. From there, we infer the unique and final state of the weights, where $A_i^* = M_i A_i = \Lambda_i A_l/\Lambda_{l-1}$. Note that each $M_i$ depends on all the other $M_j$'s, again showcasing how the local balancing algorithm leads to a unique global solution. $\square$

