# OpenReview forum: "Improving Deep Learning Speed and Performance through Synaptic Neural Balance"
_NeurIPS.cc/2024/Workshop/MLNCP — MLNCP Poster_

### Official Review · Reviewer_fu7M · 2024-09-27
**Interesting theory but application to emerging accelerators unclear**

**Rating:** 6
**Confidence:** 2

**Review:**

The submission describes the theory of neural balance, the idea that the total incoming and outgoing cost of a neuron should be equal.
While this property is normally not fulfilled in neural networks, the authors propose a regularization scheme that seeks to ensure neural balance holds. By construction, the theory is restricted to activations that are linear in both directions from zero, e.g. ReLU, a potential drawback given the popularity of ELU, SELU, or GELU functions which are slightly non-bi-linear.

The theory is interesting and the regularization shows some merit in experiments. Most experiments are conducted on standard fully connected networks (FCN) on MNIST, one experiment is done with a RNN on IMDB. The FCN MNIST experiments show that L1 regularization appears to give the strongest results, while the RNN experiment only shows marginal improvement.

Neural balance appears to yield stronger improvements in the data starved regime, leading to improvements in MNIST classification of up to 8.4%. However, the experimental section here suffers from a lack of baselines or other methods to compare against (other regularizations, dropout, data augmentation). It is not entirely clear to me if the balancing process itself incurs additional compute. If so, then this should be acknowledged and results should be compared against compute-matched baselines (e.g. augmentation). If the regularization terms are indeed just added to the overall loss function, then no (significant) additional compute cost is incurred.

The theory could be made stronger. For example, one could consider information theoretic or other perspectives to explain why and how neural-balanced networks are desirable.

My main point of contention, however, is if this submission fits within the scope of the workshop. How would, for instance, an analog accelerator benefit from neural balance? The authors should make this point quite clear. If for instance, neural balanced weight matrices are easier to quantize or more noise robust, then I see how this submission would make sense.
But otherwise the relationship to emerging AI accelerators is very unclear to me; I believe the submission should go to a different workshop or conference.

---

### Official Review · Reviewer_XZLt · 2024-10-04
**A technique to reparameterize the weights of a neural network, with experiments on MNIST and the IMDB sentiment analysis dataset**

**Rating:** 5
**Confidence:** 3

**Review:**

The paper argues that a neural network trains faster and to better accuracy when the weights of a neural network satisfy a condition called “synaptic neural balancing”.

More technically, the paper proves that, in a neural network with leaky ReLU activation functions, for each neuron (or unit), there exists a functionally equivalent configuration of weights that satisfies the so-called “synaptic neural balance” condition. Synaptic neural balance of a neuron means that the Lp norm of incoming weights is equal to the Lp norm of outgoing weights.
The proof of the theoretical result is straightforward.

Then, the paper goes on with experiments on MNIST and the IMDB sentiment analysis dataset to support the claim that neural network training is faster and more efficient when the synaptic neural balance condition is enforced.

The paper is well written and reads well.



**On the novelty**

Although I don’t know the literature on the subject very well, questions related to how to initialize and reparametrize the weights optimally to achieve faster and better training have been studied for a very long time. The method presented in this work seems rather natural, so I’m a little skeptical about the novelty of this work. In particular, I am surprised that I did not see any occurrence of the word “reparametrization” in this manuscript. There is a broad literature on the effect of reparametrization of neural networks on optimization and generalization – see for example these two papers:

Salimans, Tim, and Durk P. Kingma. "Weight normalization: A simple reparameterization to accelerate training of deep neural networks." Advances in neural information processing systems 29 (2016).

Dinh, Laurent, et al. "Sharp minima can generalize for deep nets." International Conference on Machine Learning. PMLR, 2017.



**On the effectiveness of the method**

The thesis of the manuscript does make sense, intuitively. However, the experiments performed on MNIST and the IMDB sentiment analysis dataset, are very limited in scope, so it seems difficult to extrapolate from this study only. If the synaptic neural balancing method proved to be really effective, it would have implications not just for neuromorphic computing, but also more immediately for current deep learning models trained on GPUs. The fact that the technique is not used in SOTA deep learning makes me think that the benefits (if any) of the method are not worth the burden of making the training scheme more complicated by “rebalancing” the weights.

So the question is whether there is any advantage of this synaptic neural balancing method for neuromorphic computing specifically, compared to GPU-based neural networks (?)



**Other comments**

I am not favorable to using the term “bilinear” (in “BiLU activation function”), because “bilinear” has a very specific meaning in linear algebra, different from the one used here. I’d suggest more simply “leaky ReLU with amplification”, or “generalized leaky ReLU”, or something along these lines.
Also, the term “BiLU”, first mentioned at line 72, does not seem to be defined in the main text. A pointer to Appendix C1 (where it is defined) seems to be missing.

The notation g_w(w) in Eq 2 can be a little confusing. As I understand it, the two occurrences of w refer to two different things here: the first one refers to the *specific weight*, and the second one refers to the *value* of that weight. Maybe adding a one-sentence explanation would be helpful for the reader.

---

### Decision · Program_Chairs · 2024-10-10

Accept (Poster)